# Plant-Based Diets and Phytochemicals in the Management of Diabetes Mellitus and Prevention of Its Complications: A Review

**DOI:** 10.3390/nu16213709

**Published:** 2024-10-30

**Authors:** Prawej Ansari, Joyeeta T. Khan, Suraiya Chowdhury, Alexa D. Reberio, Sandeep Kumar, Veronique Seidel, Yasser H. A. Abdel-Wahab, Peter R. Flatt

**Affiliations:** 1Comprehensive Diabetes Center, Heersink School of Medicine, University of Alabama, Birmingham (UAB), Birmingham, AL 35233, USA; 2School of Pharmacy and Public Health, Department of Pharmacy, Independent University, Bangladesh (IUB), Dhaka 1229, Bangladesh; 3Centre for Diabetes Research, School of Biomedical Sciences, Ulster University, Coleraine BT52 1SA, UK; y.abdel-wahab@ulster.ac.uk (Y.H.A.A.-W.); pr.flatt@ulster.ac.uk (P.R.F.); 4Department of Pharmaceutical Sciences, College of Pharmacy, University of Arkansas for Medical Sciences (UAMS), Little Rock, AR 72205, USA; 5Natural Products Research Laboratory, Strathclyde Institute of Pharmacy and Biomedical Sciences, University of Strathclyde, Glasgow G4 0RE, UK; veronique.seidel@strath.ac.uk

**Keywords:** diabetes, dietary adjuncts, phytoconstituents, medicinal plants, glucose, insulin

## Abstract

Diabetes mellitus (DM) is currently regarded as a global public health crisis for which lifelong treatment with conventional drugs presents limitations in terms of side effects, accessibility, and cost. Type 2 diabetes (T2DM), usually associated with obesity, is characterized by elevated blood glucose levels, hyperlipidemia, chronic inflammation, impaired β-cell function, and insulin resistance. If left untreated or when poorly controlled, DM increases the risk of vascular complications such as hypertension, nephropathy, neuropathy, and retinopathy, which can be severely debilitating or life-threatening. Plant-based foods represent a promising natural approach for the management of T2DM due to the vast array of phytochemicals they contain. Numerous epidemiological studies have highlighted the importance of a diet rich in plant-based foods (vegetables, fruits, spices, and condiments) in the prevention and management of DM. Unlike conventional medications, such natural products are widely accessible, affordable, and generally free from adverse effects. Integrating plant-derived foods into the daily diet not only helps control the hyperglycemia observed in DM but also supports weight management in obese individuals and has broad health benefits. In this review, we provide an overview of the pathogenesis and current therapeutic management of DM, with a particular focus on the promising potential of plant-based foods.

## 1. Introduction

Diabetes mellitus (DM) is a multifactorial metabolic disorder that has emerged as one of the ten leading causes of death worldwide [1]. Obesity and insulin resistance or insulin deficiency are the major players in the development of DM. If not properly managed, DM may lead to severe late-stage complications that include cerebrovascular, peripheral vascular, and ischemic heart disease, kidney failure, and retinal damage [2,3]. Four different main types of diabetes are generally recognized; Type 1 diabetes (T1DM), Type 2 diabetes (T2DM), gestational diabetes (GDM), and monogenic diabetes, the most common of which is maturity-onset diabetes of the young (MODY). T1DM and T2DM are the most familiar as they affect a very considerably larger number of patients than other types [4]. T1DM, also previously called insulin-dependent DM, is associated with defective insulin secretion as a result of the destruction of the pancreatic β-cells and is predominant in children and teenagers [5]. T2DM, which affects about 90% of all cases, was previously known as non-insulin-dependent DM. This primarily affects individuals over 40 years of age, although this is being countered increasingly in the young due to increased childhood obesity. This type is characterized by pancreatic β-cell failure, causing insulin depletion, as well as insulin resistance in organs. Individuals with T2DM tend to be obese and often have a history of gestational diabetes, polycystic ovarian syndrome, cardiovascular disease (CVD), and dyslipidemia [5,6,7,8]. GDM is associated with pancreatic β-cell dysfunction and chronic insulin resistance which can occur during pregnancy. MODY is a rare genetic type of DM that commonly emerges during adolescence or early adulthood [4].

It has been estimated that around 537 million individuals have DM worldwide and that this may rise to 783 million by 2040 [9]. Up to 95% of all diabetic individuals are reported to have obesity-related type 2 diabetes (T2DM). A logistic regression model estimated that in 110 developing countries, based on United Nations (UN) population data, there were 366 million people with diabetes, and this number is expected to rise to 552 million by 2030 [10,11,12]. In developing nations such as India, Nepal, Bhutan, China, Pakistan, and Indonesia, the occurrence of T2DM has dramatically increased in recent years. In fact, studies have reported that the number of diabetes patients in low- and middle-income countries will drastically increase over the next 19 years [12]. A recent study has also reported that in Bangladesh alone, 10–15% of the adult population has some form of prediabetes or diabetes [13,14,15]. In these countries, T2DM mostly occurs in individuals between 40 and 59 years of age [16], who often have a history of childhood obesity [17]. The common symptoms of T2DM that often precipitate diagnosis include lethargy, irritation, blurry vision, confusion, polydipsia, polyuria, polyphagia, anorexia, vomiting, dehydration, sore muscles, numb feet or hands, foot infections, delayed wound healing, kidney failure, cardiovascular diseases, coma, and, in extreme cases, death [18,19,20].

While insulin is the only therapy for T1DM, patients with T2DM rely primarily on one or more of a range of oral hypoglycemic drugs that include α–glucosidase inhibitors, metformin, sulfonylureas, meglitinides, thiazolidinediones, amylin analogs, SGLT-2 inhibitors, dipeptidyl peptidase-4 (DPP-4) inhibitors, GLP-1 mimetics, and incretin receptor dual agonists. In cases where these medicines are not effective, insulin is then administered [21,22]. Hypoglycemia has been documented as one of the most severe adverse side effects of antidiabetic treatments. Nausea, bloating, gas formation, gastrointestinal disorders, and urinary and respiratory tract infections are other commonly reported side effects [23]. The use of alternative approaches to better manage DM and its late-stage complications is becoming increasingly popular in many developing countries such as India, Bangladesh, Nepal, Pakistan, Indonesia, and China. Integrating edible plants with reputed antihyperglycemic activity, such as bitter melon, moringa, clove, turmeric, neem, black seeds, or cinnamon, to name a few, in the daily diet is an attractive option that may present fewer side effects than conventional drugs [12,24,25,26,27]. In this review, we discuss the potential of plant-based dietary habits in the management of T2DM and its complications, highlighting the pharmacological effects and phytoconstituents relevant to DM of one hundred plant species. The main objective of this review is to provide the basis for future research on the antidiabetic potential of the selected plants.

## 2. Pathophysiology of Diabetes Mellitus

The pathogenesis of T2DM has been linked to underlying genetic factors, as well as obesity caused by a sedentary lifestyle and poor dietary choices. T2DM is characterized by hyperglycemia linked to hyperlipidemia, persistent inflammation, oxidative stress, mitochondrial dysfunction, and gut dysbiosis, ultimately leading to β-cell apoptosis and insulin resistance (IR) (Figure 1) [28,29,30,31,32]. As T2DM progresses, the production of advanced glycation end products (AGEs) leads to build-up in the kidney, retina, and blood vessels, which triggers micro- and macrovascular complications [33,34].

Obese individuals tend to consume more nutrients than needed, leading to an excess of body fat and glycogen. Obesity plays a large contribution to the development of T2DM [3]. One study reported that around 85% of T2DM patients are obese [35]. Moreover, the lack of regular physical activity in T2DM patients has been linked to low circulating levels of irisin, an exercise-modulated myokine that improves glucose tolerance through physical activity [36,37,38,39,40]. In some cases, the term “diabesity” is used to describe the close link between T2DM and obesity [41]. Overnutrition also causes oxidative stress and inactivates glucose transporter-4 (GLUT4) translocation, reducing glucose uptake in cells [42]. Obese individuals are more likely to develop IR as a result of a compensatory rise in insulin production (hyperinsulinemia). IR involves impaired insulin receptor signaling in tissues such as adipose tissues, which leads to a dysregulation of insulin secretion and storage. This occurs until the pancreatic β-cells fail to adequately fulfill the demand for insulin. Hence, glucose cannot enter cells of insulin-sensitive peripheral tissues and accumulates in the blood [43,44,45,46,47,48]. In the context of diabetes mellitus (DM), chronic AMPK inhibition becomes a vicious cycle. Nutrient excess, particularly from high-fat or high-glucose diets, can impede the AMPK signaling pathway. This leads to chronic inflammation, oxidative stress, and hormonal imbalances. This impaired AMPK function further worsens insulin resistance (IR) and β-cell dysfunction, key contributors to DM. Symptoms such as polyphagia (increased hunger) then arise, promoting weight gain and fueling the progression of DM [49,50]. Hyperinsulinemia and insulin resistance can also be observed in individuals where the normal function of insulin receptors or the insulin-degrading enzyme is impaired due to genetic mutations [51,52,53].

The accumulation of lipids such as triacylglycerides (TAG), diacylglycerides (DAG), ceramides, acylcarnitine, and acyl-CoAs in obese individuals also increases the risk of IR [54,55,56,57,58]. This develops via increasing intracellular DAG levels and PKC signaling, which in turn leads to the phosphorylation of IRS-1 on serine residues, disrupting normal insulin signaling pathways. This disruption impairs the ability of insulin to stimulate glucose uptake and metabolism in tissues such as muscle, liver, and adipose tissue. Over time, IR leads to β-cell dysfunction and eventually T2DM (Figure 1) [58,59]. IR in T2DM patients has also been linked to a rise in pro-inflammatory markers such as interleukin-6 (IL-6), tumor necrosis factor-alpha (TNF-α), and C-reactive protein in the bloodstream [39,60,61]. Obesity also affects mitochondria through the generation of NADH and FADH_2_, which disrupts the electron transport chain (ETC) and increases ROS production and AGEs. ROS induce oxidative stress and hamper the function of intracellular proteins and enzymes, promoting fatty acids to form toxic intracellular lipids, reducing mitochondrial energy production, and increasing IR and β-cell damage. The increased gluconeogenesis in the liver also increases the risk of hyperglycemia and subsequent organ damage [62,63,64,65,66]. This metabolic imbalance alters the structure and composition of the extracellular matrix, leading to endothelial dysfunction and increasing the risk of atherosclerosis [67]. Finally, gut dysbiosis may also influence IR by modulating glucose metabolism. Recent studies have reported that specific changes in the gut microbiota composition can either exacerbate or ameliorate insulin sensitivity and glucose tolerance, highlighting its crucial role in DM [68,69].

Unsurprisingly, a healthy diet, regular physical activity, appropriate weight loss, and even occasional fasting can ameliorate IR, β-cell function, and insulin secretory capacity, reducing the risk of T2DM and its associated complications [70,71,72].

## 3. Complications of Diabetes Mellitus

Persistent hyperglycemia, hyperlipidemia, high levels of ROS, and pro-inflammatory mediators in the bloodstream increase the risk of macrovascular complications such as coronary heart disease (CHD), stroke, peripheral artery disease, cardiomyopathy, arrythmia, cerebrovascular disease, and atherosclerosis [73,74]. Individuals with DM and hypertension are at a higher risk of developing cerebrovascular disease, peripheral vascular disease, or early coronary artery disease (CAD) [75,76,77,78,79]. Similarly, obesity is considered to be a key risk factor for heart failure (HF), CHD, and premature mortality [80,81,82]. Hormones and other circulatory factors including adipokines, growth factors, and chemokines have been reported to aggravate CVD in T2DM patients [83,84].

Diabetic patients may also suffer from various microvascular complications including neuropathy, nephropathy, retinopathy, foot damage, Alzheimer’s disease, and hearing impairment [85]. Diabetic peripheral neuropathy, characterized by pain, ulcers, sleep deprivation, and depression, affects about half of diabetic patients worldwide [85,86,87,88,89]. Factors such as genetic predisposition, age, food intake, smoking, alcohol, and other unhealthy lifestyle habits have also been implicated in the progression of diabetic peripheral neuropathy [90]. Uncontrolled blood sugar levels damage the nerves, diminishing their ability to send signals and weakening the lining of capillaries that supply nutrients and oxygen to neurons [91,92].

T2DM has been linked to an increased risk of developing Alzheimer’s disease due to the presence of overlapping neurodegenerative markers in both diseases such as oxidative stress, inflammation, and mitochondrial dysfunction [93]. On the other hand, diabetic nephropathy, characterized by microalbuminuria, elevated blood glucose, high hemoglobin A1C (HbA1c), and hypertension, is prevalent in nearly half of T2DM individuals [94,95,96,97]. Diabetic retinopathy is another severe complication of T2DM which occurs when excess blood glucose blocks the capillaries linked with the retina. This increases the risk of eye disorders such as diabetic cataracts, macular edema, and dry eye, and may even result in blindness (Figure 2) [98,99,100,101].

## 4. Current Approaches for the Management of T2DM

A balanced diet, regular physical exercise, and the avoidance of high-calorie foods are the first approaches recommended for the management of T2DM and its complications. This is usually supplemented by the use of antidiabetic medicines to achieve optimal glycemic control and provide long-term relief from DM [99,100]. Current oral antidiabetic drugs include sulfonylureas, biguanides, thiazolidinediones, α-glucosidase inhibitors, SGLT2 inhibitors, meglitinides, DPP-IV inhibitors, and amylin analogs. Sulfonylureas bind to sulfonylurea receptors (SURs) and act by blocking ATP-sensitive K^+^-channels in the pancreatic β-cell plasma membrane, leading to the inhibition of K^+^ efflux, membrane depolarization, the opening of voltage-gated Ca^2+^ channels, an influx of Ca^2+^, and the triggering of insulin secretion by exocytosis [101,102,103]. However, sulphonylureas present adverse side effects such as hypoglycemia, increased risk of CVD, and nausea [104,105,106]. Meglitinides work in similar fashion but affect a slightly different bonding site on SURs [107]. At high doses, these agents may cause severe hypoglycemia, upper respiratory tract infections, diarrhea, and headaches [107,108]. Biguanides inhibit the mitochondrial respiratory chain in the liver, activating the AMPK pathway, enhancing insulin sensitivity, suppressing gluconeogenesis, and reducing both hepatic glucose output as well as glucose entry into the circulation from the intestines [109,110,111,112]. Although biguanides are very popular antidiabetic drugs, they still cause undesirable effects such as diarrhea, lactic acidosis, and hemolytic anemia [113,114]. Thiazolidinediones (TZDs) act by activating the gamma isoform of the peroxisome proliferator-activated receptor (PPAR-γ), increasing glucose and lipid metabolism, providing energy homeostasis, and promoting GLUT4 translocation [115]. Adverse effects associated with TZDs include weight gain, hepatotoxicity, and even bladder cancer [116]. Inhibitors of α-glucosidase decrease the intestinal activity of this enzyme, delaying carbohydrate digestion and absorption and improving hepatic lipogenesis, triglyceride levels, and postprandial glucose [117]. However, some TZDs have been discontinued due to an increased risk of cardiovascular diseases. Alongside that, their use may cause hepatitis, increased flatulence, and other gastrointestinal complications [118]. Sodium-glucose cotransporter 2 (SGLT2) inhibitors act to promote urinary glucose excretion and not only treat DM but also reduce inflammation, Na^+^/H^+^-exchange, and hyperuricemia. They elevate lysosomal degradation, autophagy, and erythropoietin levels, while also preventing ischemia [119]. Although SGLT-2 drugs are popular for alleviating diabetes, they still carry the risk of side effects, including volume depletion, increased urination, acute kidney injury, and genitourinary infections [120]. Dipeptidyl-peptidase IV (DPP-4) inhibitors increase the levels of incretin hormones, glucagon-like peptide-1 (GLP-1), and glucose-dependent insulinotropic peptide (GIP). Their side effects include urinary and upper respiratory tract infections as well as headaches [121].

In many cases, oral drugs alone are not enough to control hyperglycemia and injectable therapy is required to successfully manage DM. The most common injectable therapy is synthetic insulin. Insulin works by binding to the insulin receptor, activating a cascade of intracellular signaling events [122,123]. Although insulin is very effective in DM, it may lead to severe hypoglycemia, dizziness, sweating, palpitations, headache, blurred vision, and abdominal pain [123]. Amylin analogs, often used in combination with other antidiabetic drugs, inhibit glucagon secretion, delay gastric emptying time, and improve postprandial glycemia [124,125]. Their adverse effects include severe hypoglycemia, nausea, and weight loss [124,125,126]. GLP-1 and GIP analogs are also used as injectable therapies for DM. GLP-1 drugs stimulate insulin secretion and inhibit glucagon release from pancreatic α-cells, suppress appetite, and promote extra pancreatic activity by delaying gastric emptying. Scientists are also assuming that there might be a connection between the progression of pancreatitis and C-cell tumors; however, there is still a lack of studies related to these conditions [127,128,129,130]. GIP and GLP-1 dual agonists, such as Mounjaro, enable insulin secretion through activation of β-cell GIP receptors and appear to greatly enhance the satiety and weight loss encountered with GLP-1R activation alone, aiding obesity [129,130]. The most common side effects of these injectables are severe nausea, vomiting, and body disfiguration due to excessive weight loss [130]. An overview of the current oral and injectable antidiabetic drugs, their pharmacological actions, and adverse side effects are presented in Figure 3.

## 5. Plant-Based Diets and Their Role in the Prevention and Management of DM

A lifelong treatment with conventional antidiabetic drugs presents limitations in terms of side effects and costs. In this context, plants with antidiabetic activity have become an alternative treatment option for many patients as they are generally more accessible, less costly, and present fewer adverse side effects than manufactured drugs. They are also gaining popularity in scientific research as an attractive source for the discovery of new drug templates [131,132]. Numerous epidemiological studies have highlighted the importance of a diet rich in plant-based foods (vegetables, fruits, spices, and condiments) in the prevention and management of diseases, including DM. Plant-based foods and their beneficial constituents are often absent in the typical Western diet that predominantly features processed foods, red meat, and fast-acting carbohydrates, which contribute to the development and progression of T2DM. Dietary fiber-rich herbs and fruits, in particular, have been reported to regulate hyperglycemia and mitigate diabetic complications (Table 1) [133]. Understanding how these plant-derived constituents affect the pathophysiology of T2DM can provide a useful strategy to better prevent this disease and its complications (Figure 4). It can also reduce reliance on synthetic antidiabetic drugs [134,135,136].

For example, aloe vera, neem, holy basil, and betel leaf possess anti-inflammatory and hypoglycemic properties that help regulate blood glucose and body weight. Citrus fruits (e.g., lemon, orange, and pomelo), along with mango, apple, pineapple, and berries (e.g., strawberry, blueberry, blackberry, and mulberry), are high in fiber and antioxidants. They promote satiety and reduce oxidative stress. Stone fruits such as peach, guava, avocado, kiwi, lychee, grapes, jackfruit, dragon fruit, passion fruit, star fruit, pomegranate, papaya, fig, watermelon, plum, and java plum, as well as dates and apricots, contribute to improving metabolic health. Amla and olives contain unique phytochemicals that enhance insulin sensitivity. Tamarind, Bengal currant, cocoa, coconut, cashew nut, almond, walnut, and seeds such as chia, white sesame, black seeds, cumin, fenugreek, mustard, coriander, and nutmeg, provide essential fatty acids and micronutrients that are crucial for metabolic function [137,138,139,140,141,142,143,144,145,146,147,148,149,150,151,152,153,154,155,156,157,158,159,160,161,162,163,164,165,166,167,168,169,170,171,172,173,174,175,176,177,178,179,180,181,182,183,184,185,186,187,188,189,190,191,192,193,194,195,196,197,198,199,200,201,202,203,204,205,206,207,208,209,210,211,212,213,214,215,216,217,218,219,220,221,222,223,224,225,226,227,228,229,230,231,232,233,234,235,236,237,238,239,240,241,242,243,244,245,246,247,248,249,250,251,252,253,254,255,256,257,258,259,260,261,262,263,264,265,266,267,268,269,270,271,272,273,274,275,276,277,278,279,280,281,282,283,284,285,286,287,288,289,290,291,292,293,294,295,296,297,298,299,300,301,302,303,304,305,306,307,308,309,310,311,312,313,314,315,316,317,318,319,320,321,322,323,324,325,326,327,328,329,330,331,332,333,334,335,336,337,338,339,340,341,342,343,344,345,346,347,348,349,350,351,352,353,354,355,356,357,358,359,360,361,362,363,364,365,366,367,368,369,370,371,372,373,374,375,376,377,378,379,380,381,382,383,384,385,386,387,388,389,390,391,392,393,394,395,396,397,398,399,400,401,402,403,404,405,406,407,408,409,410,411,412,413,414,415,416,417,418,419,420,421,422,423,424,425,426,427,428,429,430,431,432,433,434,435,436,437,438,439,440,441,442,443,444,445,446,447,448,449,450,451,452,453,454,455,456,457,458,459,460,461,462,463,464,465,466,467,468,469,470,471,472,473,474,475,476,477,478,479,480,481,482,483,484,485,486,487,488,489,490,491,492,493,494,495]. Fiber-rich grains such as corn, oats, and quinoa, as well as legumes including chickpea, pea, kidney bean, mung bean, and soya bean, help maintain steady blood glucose levels and manage obesity. Vegetables such as bitter gourd, snake gourd, ridge gourd, bottle gourd, sweet potato, moringa, okra, taro, asparagus, eggplant, beetroot, pumpkin, cabbage, broccoli, radish, carrot, tomato, cucumber, lettuce, spinach, centella leaves, and mushrooms are excellent for their low-calorie, high-nutrient profiles. Herbs and spices such as mint, parsley, celery, rosemary, oregano, curry leaves, bay leaves, clove, saffron, cinnamon, red pepper, turmeric, ginger, and garlic enhance the metabolic rate and have antidiabetic effects. Onions, tea, coffee, China rose, and vinca rosea also contribute to improving glucose metabolism and controlling body weight. The incorporation of these foods into a balanced diet can support the management of T2DM and obesity by promoting better glycemic control, enhancing insulin sensitivity, and helping with weight loss (Table 2) [410,411,412,413,414,415,416,417,418,419,420,421,422,423,424,425,426,427,428,429,430,431,432,433,434,435,436,437,438,439,440,441,442,443,444,445,446,447,448,449,450,451,452,453,454,455,456,457,458,459,460,461,462,463,464,465,466,467,468,469,470,471,472,473,474,475,476,477,478,479,480,481,482,483,484,485,486,487,488,489,490,491,492,493,494,495,496,497,498,499,500,501,502,503,504,505,506,507,508,509,510,511,512,513,514,515,516,517,518,519,520,521,522,523,524,525,526,527,528,529,530,531,532,533,534,535,536,537,538,539,540,541,542,543,544,545,546,547,548,549,550,551,552,553,554,555,556,557,558,559,560,561,562,563,564,565,566,567,568,569,570,571,572,573,574,575,576,577,578,579,580,581,582,583,584,585,586,587,588,589,590,591,592,593,594,595,596,597,598,599,600,601,602,603,604,605,606,607,608,609,610,611,612].

## 6. Plant-Based Diets, Edible Plants, Dietary Adjuncts, and Their Phytochemicals for the Management of DM and Prevention of DM Complications

Medicinal plant-based diets, rich in bioactive compounds, have gained attention for their potential in preventing and managing chronic diseases, including diabetes. Phytochemicals such as flavonoids, anthocyanins, carotenoids, saponins, tannins, and polyphenols, found in a variety of plant-based foods such as fruits, vegetables, legumes, and whole grains, play an essential role in managing diabetes. These compounds help regulate blood sugar through a possible multitude of actions, including enhancing insulin sensitivity or secretion, inhibiting enzymes that break down carbohydrates, and reducing glucose production in the liver. They also improve gut health, trigger the release of glucoregulatory or satiating gut hormones, reduce inflammation, and combat oxidative stress, all of which contribute to better overall metabolic function and diabetes prevention (Figure 5) [137].

There are numerous studies conducted on medicinal plant-based diets that report modulation of antidiabetic parameters. For example, okra is rich in flavonoids and it has the ability to enhance GLUT-4 translocation and inhibit carbohydrate digestive enzymes to slow the rate that glucose appears in the blood stream [138]. Pineapple, on the other hand, is rich in polyphenols that exert anti-inflammatory effects through the reduction of ROS and oxidative stress [176]. Furthermore, tea is high in tannin, which may also contribute to its insulin sensitizing and anti-inflammatory actions [228]. Similarly, oats, which are a common staple food and abundant in saponin, have the ability to improve the lipid profile and insulin resistance [201]. Saffron is also enriched with beneficial phytochemicals such as flavonoids and carotenoids, which contribute to its potent antidiabetic actions such as the reduction of cholesterol, inflammatory cytokines, and mitochondrial dysfunction [302,314,315]. Polyphenols, found in fruits such as apples and grapes, can manage diabetes by activating the AMPK pathway to improve insulin sensitivity, inhibiting α-amylase and α-glucosidase to slow carbohydrate digestion, as well as by enhancing the PI3K/Akt pathway for enhanced glucose uptake [396,602]. Additionally, anthocyanins are common antidiabetic phytochemicals present in berries and purple diets that can help to regulate blood sugar by boosting insulin sensitivity via the PI3K/Akt pathway, protect β-cells through the Nrf2 antioxidant pathway, and inhibit carbohydrate-digesting enzymes to prevent post-meal glucose spikes [345,501,512]. Accordingly, phytochemicals present in plant-based diets can provide significant benefits in diabetes management by enhancing insulin sensitivity, reducing oxidative stress, and regulating blood sugar levels through modulating numerous antidiabetic actions, making them a promising natural adjunct to traditional therapies for improving metabolic health. A pharmacological summary of the actions of widely used medicinal plant-based diets is illustrated in Figure 5.


***Abelmoschus esculentus* L. (Okra)**


*Abelmoschus esculentus* L. (Malvaceae), known as okra, is a nutritious vegetable that is also used as a remedy for chronic kidney disease, T2DM, and cardiovascular and hypertensive diseases [137]. The highly nutritious okra fruit contains oxalic acid, pectin, flavonoids, D-galactose, L-rhamnose, and D-galacturonic acid, which are reported to inhibit α-amylase and α-glucosidase enzymes and increase GLUT-4 translocation [138,139].

2.
***Actinidia chinensis* (Kiwi)**


*Actinidia chinensis* or kiwi (Actinidiaceae) is a beneficial fruit for dyspepsia, vomiting, loss of appetite, and diabetes [140]. Kiwi lowers cholesterol, LDL, fasting plasma glucose, and postprandial glucose levels. It has also been reported to reduce body weight and inhibit the release of pro-inflammatory cytokines such as interleukin-1(IL-1) and IL-6 in T2DM patients [141]. Kiwi also regulates superoxide dismutase (SOD) and glutathione levels. It inhibits the activity of alanine aminotransferase (ALT) and aspartate aminotransferase (AST), two enzymes associated with insulin resistance and metabolic syndrome. Kiwi also improves serum microRNA-424, nuclear factor erythroid 2–related factor 2 (Nrf2), and Kelch-like ECH-associated protein 1 (Keap1), and the dysregulation of these markers may exacerbate oxidative stress, inflammation, and disease progression [142]. Kiwi is rich in triterpenoids, polyphenols, amino acids, and minerals that may exert antidiabetic activity owing to hypolipidemic, anti-inflammatory, antioxidant, and antihyperglycemic properties [143].

3.
***Aegle marmelos* (Stone apple)**


*Aegle marmelos*, also called stone apple/golden apple/bael, is a plant from the Rutaceae family traditionally used for inflammation, asthma, hyperglycemia, colitis, flatulence, dysentery, fever, pain, and hepatitis and fungal infections [144]. Recent studies have indicated that it improves insulin production, inhibits glucose absorption and α-amylase activity, and lowers blood glucose levels [145]. Some of its phytochemicals, namely *p*-cymene, oleic acid, linolenic acid, myristic acid, and retinoic acid, have antidiabetic, cardioprotective, antioxidant, and anti-inflammatory properties [146].

4.
***Agaricus bisporus* (Mushroom)**


*Agaricus bisporus* (Agaricaceae) is familiarly known as the button mushroom. It is a valuable ethnomedicine for diabetes, coughs, influenza, asthma, cancer, and hepatic disorders [147,148]. Mushrooms have numerous health benefits, with antioxidant, immunoboosting, anticholesterolemic, antitumor, and antibacterial properties. They boost natural killer cells to fight infections and tumors. The presence of lectins, β-glucans, polyphenols, *p*-hydroxybenzoic acid, protocatechuic acid, agllic acid, cinnamic acid, *p*-coumaric acid, ferulic acid, chlorogenic acid, and catechin in mushrooms improves hyperglycemia by regulating insulin and glucagon secretion [149,150,151].

5.
***Allium cepa* (Onion)**


*Allium cepa* (Amaryllidaceae) or onion has been used as a treatment for wounds, scars, keloids, bee stings, dysmenorrhea, vertigo, fainting, migraine, bruises, earache, jaundice, pimples, and diabetes [152]. Onion significantly decreases α-glucosidase activity and oxidative stress, boosts insulin secretion, and protects pancreatic β-cells [153]. Onion has numerous health benefits beyond its antidiabetic properties, as it also boasts antioxidant, analgesic, antimicrobial, anti-inflammatory, and immune-boosting activity. The presence of quercetin, apigenin, rutin, myricetin, kaempferol, catechin, resveratrol, and anthocyanins may contribute to its glucose- and cholesterol-lowering effects [154,155,156].

6.
***Allium sativum* L. (Garlic)**


*Allium sativum* L. (Amaryllidaceae) or garlic is a popular folk medicine for flu, hypertension, high cholesterol, cancer, cardiovascular disease, diarrhea, preeclampsia, arthritis, diabetes, and kidney stones [157]. Garlic lowers plasma glucose levels, enhances insulin production and insulin secretion, improves glucose tolerance and insulin sensitivity, and increases GLUT4 expression [158,159]. Garlic is rich in organosulfur phytoconstituents such as ajoene, cysteine, and allicin, as well as β-resorcylic acid, gallic acid, rutin, quercetin, and protocatechuic acid, which exhibit antioxidant, renoprotective, and antihyperglycemic effects. Allicin and quercetin play crucial roles in enhancing insulin sensitivity and improving glucose uptake [160,161,162].

7.
***Aloe barbadensis Mill.* (Aloe vera)**


*Aloe barbadensis Mill.* (Asphodelaceae) has a long history as an ethnomedicine for wounds, constipation, skin diseases, colic, worm infestations, hypertension, and diabetes [163,164]. Aloe vera improves insulin resistance, body weight, and prediabetic conditions via the inhibition of fructosamine, carbonyl protein, and the formation of AGEs such as N*^ɛ^*-(carboxymethyl) lysine (CML). It also has *α*-amylase and *α*-glucosidase inhibitory activity [165,166]. It also reduces fasting and postprandial blood glucose, triglycerides, and total cholesterol levels. The antidiabetic properties of aloe vera have been attributed to the presence of flavonoids, arginine, and phenolic acids [164,166,167,168].

8.
***Anacardium occidentale* L. (Cashew nuts)**


*Anacardium occidentale* L. (Anacardiaceae), also called cashew nut, has medicinal value in alleviating fevers, aches, pains, diarrhea, diabetes, skin irritation, and arthritis [169]. Cashew nut is reported to decrease hepatic gluconeogenesis, a process in the liver that produces glucose. This helps lower blood sugar levels [170]. Studies suggest that specific amino acids (e.g., arginine and isoleucine) and fatty acids (e.g., arachidic acid) found in cashew nuts, along with other compounds such as cyanidin and peonidin, may play a role in the activity of cashew nuts by enhancing insulin sensitivity and reducing oxidative stress and blood glucose [171,173]. Anacardic acids, also present in cashew nuts, may have a potential role in mitigating diabetic complications as they possess anti-cytotoxic (protecting cells), antimicrobial, and antibacterial effects. [172].

9.
***Ananas comosus* (Pineapple)**


*Ananas comosus* (Bromeliaceae), also known as pineapple, is traditionally used as a remedy for pain, skin diseases, edema, wound, indigestion, diabetes, and blood clotting [173,174,175]. Pineapple leaves, peels, and pulp can lower blood sugar and glycated albumin levels, reduce body weight, increase insulin secretion, and increase high-density lipoprotein (HDL) cholesterol levels by inhibiting HMG-CoA reductase and activating lipoprotein lipase (LPL) [176,177,178]. Bromelain, one of the phytoconstituents of pineapple, has anti-inflammatory, hypoglycemic, anticoagulant, and antioxidant activities [179].

10.***Apium graveolens*** L. (Celery)

*Apium graveolens* L. (Umbelliferrae) or celery is useful for arthritis, spleen dysfunction, diabetes, sleep disturbances, and CNS disorders [180]. This food source helps maintain healthy blood sugar levels by enhancing insulin sensitivity and promoting the translocation of GLUT4 receptors to the cell surface, followed by enhancing glucose uptake into muscle. This, in turn, can improve mitochondrial function and reduce inflammation [181,182,183]. Celery is rich in quercetin, thymoquinone, coumaric acid, and gallic acid, with anti-inflammatory, anticoagulant, hypolipidemic, hepatoprotective, and neuroprotective properties [184,185].

11.
***Artocarpus heterophyllus* (Jackfruit)**


*Artocarpus heterophyllus* (Moraceae) or jackfruit is a traditional remedy for wounds, cancer, and diabetes [186,187]. Its fruit, bark, seeds, leaves, and roots all have antidiabetic properties [188,189,190]. Studies have reported that jackfruit significantly ameliorates body weight, lipid profile, abnormal hematological parameters, creatine, bilirubin, and urea levels, and reduces albumin levels in diabetic rats. It also has inhibitory activity against α-amylase and α-glucosidase enzymes and can improve the lipid profile (i.e., LDL and HDL cholesterol) and fasting and blood glucose levels [191,192]. Phytochemicals such as carotenoids, tannins, volatile acids, sterols, chrysin, isoquercetin, and silymarin contribute to the pharmacological properties of jackfruit [192].

12.
***Asparagus officinalis* (Asparagus)**


*Asparagus officinalis* (Asparagaceae), known as asparagus, is a remedy for diabetes, asthma, rheumatism, and liver and kidney diseases [193]. Recent studies suggest that it enhances insulin secretion and β-cell function in a rat model of T2DM [194]. Asparagus elicits its hypoglycemic properties by significantly lowering fasting blood glucose, hepatic glycogen, and triglyceride levels, as well as reducing body weight [195]. Asparagine, tyrosine, arginine, saponins, resin, and tannins are the main active phytoconstituents of asparagus. Among them, saponins are the main constituent that contributes to its hypoglycemic effects, as well as its antibacterial, anti-inflammatory, antioxidant, antidiarrheal, and anticarcinogenic properties [196,197].

13.
***Avena sativa* (Oats)**


*Avena sativa* (Poaceae) or oats are a popular breakfast meal. Oats are also a remedy for dermatitis, cancer, diabetes, and cardiovascular disease [198]. One study found that the continuous consumption of oatmeal cookies led to significant improvements in blood glucose levels and plasma insulin in diabetic rats [199]. β-glucan, oleic acid, linoleic acid, caffeic acid, coumaric acid, gallic acid, and avenanthramides are the active phytoconstituents of oats. They lower glycosylated HbA1c, fasting and postprandial blood glucose, and total cholesterol and LDL cholesterol levels, as well as improving insulin resistance in diabetic patients [199,200]. β-glucan is the major component of oats that reduces blood glucose and helps with losing weight [201,202].

14.
***Averrhoa carambola* L. (Star fruit)**


*Averrhoa carambola* L. (Oxalidaceae) is commercially known as star fruit. It is abundantly consumed in tropical and subtropical countries where it is also traditionally used for chronic headache, fever, cough, gastroenteritis, diarrhea, diabetes, skin inflammation, hypertension, and hyperglycemia [203,204,205]. Catechin, epicatechin, procyanidins, gallic acid, protocatechuic acid, ferulic acid, rutin, isoquercitrin, quercitrin, C-glycosides, leucoanthocyanidins, and triterpenoids in star fruit modulate insulin secretion, glucose uptake, and glycogen synthesis [206,207].

15.
***Azadirachta indica* (Neem)**


*Azadirachta indica*, known as neem, is a plant from the Meliacae family that is used to cure fever, skin ailments, infection, inflammation, diabetes, and dental ailments [208,209]. Its leaves, stem, bark, and seed oil have been reported to control glycemia, improve endothelial dysfunction, reduce systemic inflammation, enhance glucose transporter 4 (GLUT-4) translocation, and inhibit α-glucosidase. The antidiabetic effects of this plant are likely to be due to the presence of phytoconstituents such as nimbidin, nimbin, nimbidol, quercetin, and nimbosterone [210,211,212].

16.
***Beta vulgaris* (Beetroot)**


*Beta vulgaris* (Chenopodiaceae) or beetroot is a traditional cure for diabetes, loss of libido, stomachaches, arthritis, and constipation [213]. Beetroot shows antidiabetic activity by inhibiting gluconeogenesis, glycogenolysis, and α-amylase and α-glucosidase. It is rich in lycopene, betalains such as betanin, the flavonoids betagarin, betavulgarin, quercetin, and kaempferol, carotenoids, and coumarins. Among them, betanin is the main constituent that can mitigate diabetic complications [214,215].

17.
***Brassica juncea* (Mustard)**


*Brassica juncea* (Brassicaceae), known as mustard, is an effective remedy for arthritis, footache, lumbago, diabetes, and rheumatism [216,217]. Mustard has been reported to control blood sugar levels in people with diabetes by enhancing insulin secretion, improving the utilization of glucose, and reducing glucose absorption from the gut. These effects can be attributed to several beneficial phytochemicals including chlorogenic acid, kaempferol and other flavonoids, sinigrin, *p*-coumaric acid, vanillic acid, polyphenols, allyl isothiocyanate, cinnamic acid, and aniline [218,219].

18.
***Brassica oleracea* var. capitata (Cabbage)**


*Brassica oleracea* var. capitata or cabbage is a member of the Brassicaceae family. Cabbage is traditionally used to prevent injuries, gastritis, peptic ulcers, irritable bowel syndrome, diabetes, and idiopathic cephalalgia [220]. It shows antihyperglycemic activity via enhancing peripheral insulin sensitivity and insulin production by pancreatic β-cells. This has been attributed to the presence of myricetin, quercetin, kaempferol, apigenin, luteolin, glycitein, biochanin A, and formononetin [221,222].

19.
***Brassica oleracea* var. italica (Broccoli)**


*Brassica oleracea* var. italica (broccoli) is a vegetable from the Brassicaceae family that is well-known for its antioxidant, antimicrobial, anti-inflammatory, antihyperglycemic, and antitumor properties [223]. Broccoli increases insulin sensitivity, reduces glucose production, and inhibits ROS formation and the activity of α-amylase and α-glucosidase, contributing to lowering hyperglycemia [223,224]. Glucosinolates, isothiocyanates, sulforaphane, sinapic acid, gallic acid, chlorogenic acid, apigenin, kaempferol, luteolin, quercetin, and myricetin are the major phytochemicals found in broccoli that help to manage diabetes by improving insulin sensitivity, reducing inflammation, and combating oxidative stress. They also regulate glucose metabolism and protect pancreatic β-cells [224].

20.
***Camellia sinensis* L. (Tea)**


*Camellia sinensis* L., or tea, from the Theaceae family is a plant widely consumed as a beverage. It is also a reputed remedy for flatulence, indigestion, vomiting, diarrhea, hyperglycemia, and stomach discomfort [225,226]. Tea alleviates diabetic complications via the suppression of insulin resistance, reduction of oxidative stress, inhibition of α-amylase and α-glucosidase activity, and regulation of cytokine production. It also enhances insulin secretion and glucose tolerance, as well as inhibiting glycation and the activity of dipeptidyl peptidase-4 (DPP-IV) [225,226,227]. Tea is a rich source of bioactive compounds, including theophylline, theanine, proanthocyanidins, caffeine, myricetin, kaempferol, quercetin, chlorogenic acid, coumarylquinic acid, theogallin, catechin, and epicatechin, which exhibit antidiabetic activity by enhancing insulin sensitivity, regulating glucose metabolism, reducing oxidative stress, and improving pancreatic β-cell function [228].

21.
***Capsicum annuum* L. (Red pepper)**


*Capsicum annuum* L. (Solanaceae), identified as red pepper, is an ethnomedicine for dyspepsia, ulcer, anorexia, gastrointestinal disorders, and diabetes [229]. Recent studies reported that it exhibits glucose-lowering action via inhibition of gluconeogenesis, activation of AMPK, and stimulation of both GLUT-4 translocation and glucose uptake in skeletal muscles of obese diabetic rats [230,231]. These effects may be attributable to a rich content of carotenoids and flavonoids such as apigenin, quercetin, and isoquercetin. Red pepper has a range of other health benefits, including scavenging free radicals (antioxidant effect), promoting healthy weight management, reducing inflammation, and even potentially offering anticancer properties [232,233].

22.
***Carica papaya* (Papaya)**


*Carica papaya* (Caricaseae), commonly called papaya, has been used for centuries to treat high blood pressure, dengue, obesity, jaundice, respiratory diseases, malaria, diabetes, and wounds [234,235]. Papaya contains phytomolecules such as papain, quercetin, kaempferol, *p*-coumaric acid, β-carotene, linalool, oleic acid, tannins, saponins, and α-tocopherol, which can inhibit α-amylase and α-glucosidase activity as well as lower oxidative stress and plasma blood glucose levels [236,237].

23.
***Carissa carandas* (Bengal currant)**


*Carissa carandas* (Apocynaceae), known as koromcha or Bengal currant, is a remedy for asthma, constipation, diarrhea, diabetes, malaria, myopathic spams, fever, epilepsy, and seizures [238]. Recent studies suggest that Bengal currant significantly reduces diabetes-induced inflammation and lowers blood glucose levels via inhibition of α-amylase and α-glucosidase [239,240,241,242]. Lignans, flavonoids, steroids, phenolic acids, and alkaloids present in Bengal currant have anti-inflammatory, antibacterial, antifungal, antioxidant, and hepatoprotective effects. Lignans regulate blood glucose levels and oxidative stress [241].

24.
***Catharanthus roseus* L. (*Vinca rosea*)**


*Catharanthus roseus* L. (Apocyanaceae), also known as Vinca rosea, is a plant popularly used to treat cancer, diabetes, stomach disorders, and kidney, liver, and cardiovascular disorders [243,244]. It is reported to exert its antidiabetic effect through increasing β-cell-mediated insulin secretion via effects on Ca^2+^ channels. It was also shown to enhance glucose metabolism, protect pancreatic β-cells from oxidative stress, and improve insulin sensitivity. Gallic acid, rutin, coumaric *p* acid, caffeic acid, quercetin, kaempferol, chlorogenic acid, ellagic acid, and coumarins are thought to be responsible for the anti-hyperglycemic properties of this plant. The presence of alkaloids in *C. roseus* has also been reported to improve insulin secretion from β-cells [245,246,247].

25.
***Centella asiatica* L. (Centella leaves)**


*Centella asiatica* L. (Apiaceae), referred to as centella leaves, is an excellent ethnomedicine for leprosy, lupus, ulcers, eczema, psoriasis, diarrhea, fever, diabetes, and anxiety [248]. Centella blocks ATP-sensitive K^+^ channels to enhance insulin secretion and control hyperglycemia [249]. According to recent studies, it reduces oxidative stress and inflammation in diabetic patients. Some active phytoconstituents in centella leaves include triterpenes (asiaticoside, madecassic acid, and madecassoside), centellase, flavonoids (quercetin and kaempferol), phytosterols (campesterol, sitosterol, and stigmasterol), ferulic acid, and chlorogenic acid [250,251].

26.
***Chenopodium quinoa* (Quinoa)**


*Chenopodium quinoa* (Amaranthaceae), or quinoa, is a gluten-free high-protein cereal reported to ameliorates dyslipidemia, diabetes, and heart disease [252]. It is regarded as a ‘functional food’ as it contains a high amount of essential amino acids, fatty acids, vitamins, minerals, and dietary fibers [253,254]. Phytosterols, phytoecdysteroids, phenolics, tocophenols, betalains, tannins, and glycine betaine are the beneficial phytochemicals in quinoa that elicit both antidiabetic and anti-obesity effects by inhibiting α-glucosidase, regulating body weight, improving insulin sensitivity, and reducing postprandial glycemia and lipid accumulation in skeletal muscle [255,256,257,258].

27.
***Cicer arietinum* L. (Chickpea)**


*Cicer arietinum* L. (Fabaceae) commonly known as chickpea, is a reputed cure for digestive disorders, cancer, cardiovascular disease, and diabetes because of its high dietary fiber content. Recent findings recognized it as a healthy food staple that exerts hypoglycemic activity via inhibiting α-amylase, α-glucosidase, and dipeptidyl-4 (DPP4) enzymes. Chickpea has high antioxidant properties and inhibits the enzymes associated with carbohydrate metabolism [259,260,261]. It is rich in unsaturated fatty acids that help lower blood cholesterol levels and reduce inflammation and weight gain [262]. Its phytoconstituents, including uridine, adenosine, tryptophan, 3-hydroxy-olean-ene, and biochanin, contribute to its antihypertensive, antioxidant, hypocholesterolemic, and anticancer effects [263,264].

28.
***Cinnamomum verum* (Cinnamon)**


*Cinnamomum verum* (Lauraceae), also known as cinnamon, is an ethnomedicine used for diabetes, nausea, vomiting, flatulence, fever, halitosis, arthritis, coughing, hoarseness, impotence, frigidity, cephalalgia, odontalgia, and cardiac and urinary disorders [265]. Cinnamon exerts its antihyperglycemic effects by increasing GLUT-4 translocation in insulin-sensitive tissues, upregulating mitochondrial UCP-1, inhibiting α-glucosidase, and stimulating insulin secretion [266,267]. Its phytoconstituents, including cinnamaldehyde, cinnamates, cinnamic acid, eugenol, cinnamyl acetate, β-sitosterol, flavonoids, glucosides, coumarins, vanillic acid, and syringic acid, have antihyperglycemic and anti-inflammatory properties [265,268].

29.
***Citrullus lanatus* (Watermelon)**


*Citrullus lanatus* (Cucurbitaceae), or watermelon, is a fruit traditionally used to treat gastrointestinal disorders, urinary infections, fever, constipation, and emetic problems [269,270]. It improves glucose transporter (GLUT 2 and GLUT 4) levels and suppresses oxidative stress, as well as α-glucosidase and α-amylase activity. Some of the phytoconstituents of watermelon that may contribute to its pharmacological action include stigmasterol, rutin, *p*-coumaric acid, quercetin, kaempferol, β-carotene, and α-tocopherol [271,272].

30.
***Citrus limon* (Lemon)**


*Citrus limon (Rutaceae*), also known as lemon, is a common ethnomedicine used for coughs, scurvy, colds, hypertension, fever, rheumatism, sore throats, diabetes, irregular menstruation, and liver diseases [273,274,275]. Lemon exerts antihyperglycemic activity by increasing insulin sensitivity, GLUT4 translocation, and glucose uptake, inhibiting α-glucosidase, protein tyrosine phosphatase, and aldose reductase, and reducing the formation of AGE products [276,277,278]. Previous studies have shown that it reduces plasma glucose levels, as well as LDL, VLDL, total cholesterol, triglyceride, free fatty acid, and phospholipid levels. Its bioactive constituents include limocitrin, D-limonene, hesperidin, and naringenin [278,279].

31.
***Citrus maxima* (Pomelo)**


*Citrus maxima* (Rutaceae), also called pomelo, is a fruit with a great ethnomedicinal value in treating asthma, fever, ulcers, diarrhea, coughs, Alzheimer’s disease, diabetes, and insomnia [280]. Pomelo has α-amylase and α-glucosidase inhibitory activity. It also inhibits the angiotensin I converting enzyme, which notably lowers blood glucose levels and improves diabetic complications [281]. Pomelo possesses antioxidant, anti-inflammatory, anti-obesity, and hypolipidemic properties in addition to its hypoglycemic effects due to the presence of amino acids, terpenoids, sterols, carotenoids, and polyphenols [281,282,283].

32.
***Citrus reticulata* (Orange)**


*Citrus reticulata*, also known as orange, is a plant from the Rutaceae family that has been shown to be beneficial in the treatment of Alzheimer’s disease, coughs, phlegm, diabetes, hepatic steatosis, and cancer [284,285,286]. Orange increases the expression of GLUT-4 and the β-subunit insulin receptor, which further helps with insulin sensitivity [287,288,289]. Orange peel contains flavonoids such as hesperidin and naringenin that have antihyperglycemic, antihyperlipidemic, anti-obesity, and antioxidant properties [287,288].

33.
***Cocos nucifera* (Coconut)**


*Cocos nucifera,* or coconut, is an important species from the Arecaceae family, commonly used as a folk remedy for diarrhea, diabetes, renal diseases, stomachaches, fever, asthma, and sexually transmitted diseases [290,291,292,293]. Coconut has been reported to regenerate pancreatic β-cells, enhance metabolism in adipose tissue, and mitigate insulin resistance, hyperglycemia, dyslipidemia, inflammation, and oxidative stress [293,294,295]. It has also been shown to scavenge free radicals, inhibit α-amylase and α-glucosidase activity, and ameliorate diabetic complications, including diabetic neuropathy in streptozotocin-induced diabetic rats [297]. Coconut is rich in amino acids, fibers, tannins, resins, flavonoids, and alkaloids, which may contribute to its insulin-releasing and antihyperglycemic effects [293,294,295,296].

34.
***Coffea Arabica* L. (Coffee)**


*Coffea Arabica* L. (Rubiaceae) or coffee is another popular health drink. It is also a traditional remedy for flu, anemia, diarrhea, intestinal pain, migraines, headaches, fever, purulent wounds, pharyngitis, diabetes, and stomatitis [298]. Coffee exerts antidiabetic effects by improving insulin sensitivity, enhancing glucose metabolism, protecting pancreatic β-cells, and reducing the risk of T2DM development. It contains caffeine, chlorogenic acids (CGAs), caffeic, *p*-coumaric, vanillic, ferulic, protocatechuic acids, coffeasterin, kaempferol, quercetin, sinapic, quinolic, tannic, pyrogallic acids, trigonelline, caffeoylquinic, and dicaffeoylquinic, which substantially mitigate hyperglycemia and α-glucosidase activity and enhance insulin secretion [298,299,300].

35.
***Colocasia esculenta* (Taro)**


*Colocasia esculenta* (Araceae), or taro, is a remedy for rheumatic pain, diabetes, hypertension, and pulmonary congestion [301]. It can improve diabetic complications by decreasing blood glucose levels and reducing body weight in T2DM patients [302]. Taro contains vitexin, isovitexin, orientin, isoorientin, rosmarinic acid, and luteolin, which help to reduce blood glucose, inflammation, and oxidative stress in diabetic patients [303,304,305].

36.
***Coriandrum sativum* (Coriander)**


*Coriandrum sativum* (Apiaceae), known as coriander, is a common garnishing herb and a useful traditional remedy for diarrhea, flatulence, colic, indigestion, gastrointestinal diseases, and diabetes [306]. Coriander is helpful in the management of diabetes as it regenerates pancreatic β cells and improves their function. It also inhibits α-glucosidase, thereby slowing the digestion of complex carbohydrates [306,307,308,309,310]. Moreover, coriander plays a useful role in the management of diabetic complications, particularly alleviating diabetic nephropathy and neuropathy through the inhibition of AGE formation, inhibition of TNF-α release, and reduction of oxidative stress [307,308]. Coriander is rich in flavonoids, tocotrienols, tocopherols, sterols, and carotenoids, with antidiabetic, antioxidant, anti-obesity, and anticancer effects [309,310].

37.
***Crocus sativus* L. (Saffron)**


*Crocus sativus* L. (Iridaceae), or saffron, is a popular food additive as well as an effective remedy for central nervous system disorders and for diabetes [311,312]. Saffron is documented to improve insulin sensitivity, enhance glucose uptake, inhibit gluconeogenesis, and mitigate against oxidative stress, thereby offering a range of antidiabetic benefits. Bioactive constituents of saffron are β carotenes, crocetin, crocin, picrocrocin, zeaxanthene, and safranal. These exert their glycemic effects via α-glucosidase and α-amylase inhibitory activity [311,312,313]. Crocin, the main bioactive constituent of saffron, reduces blood glucose, LDL, cholesterol, and triglycerides levels. It also inhibits the release of pro-inflammatory cytokines and elevates glutathione levels [302,314,315].

38.
***Cuminum cyminum* L. (Cumin seeds)**


*Cuminum cyminum* L. (Apiaceae), referred to as cumin, is used as a remedy for diarrhea, dyspepsia, epilepsy, toothache, whooping cough, flatulence, indigestion, diabetes, and jaundice [316]. Cumin has been reported to enhance insulin secretion from pancreatic β-cells, improve insulin sensitivity in peripheral tissues by activating insulin signaling, regulate glucose uptake by enhancing GLUT4 translocation, and modulate key enzymes involved in glucose metabolism [316,317,318]. Cumin seeds are rich in compounds such as cuminaldehyde, safranal, and terpenes (including carvone, carvacrol, limonene, and linalool). These are believed to improve blood sugar levels by increasing pancreatic insulin and protecting insulin-producing β-cells from damage [317,318].

39.
***Cucumis sativus* L. (Cucumber)**


*Cucumis sativus* L. (Cucurbitaceae), known as cucumber, is a vegetable low in calories and with a high water content that is typically served as a salad. It is useful in treating sunburn, skin irritation, constipation, thermoplegia, gallbladder stones, hyperdipsia, and diabetes [319,320]. It also exhibits antihyperlipidemic, antioxidant, analgesic, and free radical scavenging effects [575,579]. It is a good source of cucurbitacins, cucumerin A and B, cucumegastigmanes I and II, and flavonoids such as vitexin, orientin, apigenin, and isoscoparin, which can synergistically improve plasma glucose, glycolysis, insulin sensitivity, and body weight in diabetes patients [319,321,322]. Other studies reveal that cucumber may suppress glucagon secretion and gluconeogenesis [323].

40.
***Cucurbita pepo* L. (Pumpkin)**


*Cucurbita pepo* L. (Cucurbitaceae), known as pumpkin, is a popular vegetable and folk medicine for dermatitis, depression, irritable bladder, intestinal inflammation, prostate enlargement, and hyperglycemia [324,325]. Pumpkin seeds have been reported to lower plasma and urine glucose, as well as triglycerides levels, and increase glutathione levels through upregulation of Nrf2 and P13K levels in T2DM mice [330,331,332]. Among the constituents of pumpkin seeds, flavonoids, alkaloids, polysaccharides, and polyphenols have been reported to enhance insulin secretion. The high content of carotenoids, zeaxanthin, and lutein has been implicated with improving insulin sensitivity, reducing inflammation, and protecting against oxidative stress [324,325,326,327].

41.
***Curcuma longa* L. (Turmeric)**


*Curcuma longa* L. (Zingiberaceae), commonly referred to as turmeric, is known as an extremely powerful healing agent and aid for coughs, diabetes, arthritis, gallbladder stones, dermatitis, cancer, and intestinal and gastric diseases [328]. Turmeric has multiple reputed health benefits as an antioxidant, anti-inflammatory, hepatoprotective, nephroprotective, neuroprotective, and immunomodulatory agent. A recent study reported that the ingestion of turmeric improved insulin secretion and insulin sensitivity and decreased insulin resistance [329,330,331,332]. The presence of caffeic acid, curdione, *p*-coumaric acid, demethoxycurcumin, isorhamnetin, valoneic acid, eugenol, isoshyobunone, and corymbolone in turmeric may contribute to these antidiabetic properties. Furthermore, turmeric is rich in curcumin, which induces glucose uptake and GLUT2 activity as well as notably promoting insulin production [330,331,332].

42.
***Daucus carota* (Carrot)**


*Daucus carota* (Apiaceae), widely known as carrot, is traditionally used for diarrhea, constipation, intestinal inflammation, weakness, illness, diabetes, and rickets [333]. Carrot has been reported to inhibit glucose absorption by significantly inhibiting α-glucosidase and α-amylase activity, improving insulin resistance in diabetic patients [334]. Carotenoids such as α and β-carotene are the main phytochemicals in carrot. It also contains polyacetylenes, ascorbic acid, lutein, lycopene, and anthocyanins, which can enhance insulin sensitivity and pancreatic β-cell function [335,336].

43.
***Ficus carica* (Fig)**


The fig plant, *Ficus carica,* belongs to the Moraceae family. It is a useful remedy for dermatitis, anemia, diabetes, paralysis, urinary tract infections, ulcers, and liver diseases [337]. Its leaves, pulp, stem, and root decrease body weight, LDL and VLDL cholesterol, triglycerides, and postprandial glucose levels, as well as inhibiting pancreatic β-cell apoptosis via the pancreatic AMPK, C-Jun *N*-terminal kinase, p-JNK, and caspase-3 pathways [338,339]. The fruit is rich in eugenol, anthocyanins, phenolic acids, flavones, and flavanols which may be responsible for the antimicrobial, neuroprotective, antioxidant, and anti-inflammatory properties of this plant [340,341,342].

44.
***Fragaria ananassa* (Strawberry)**


*Fragaria ananassa* (Rosaceae), known as strawberry, is an effective remedy for wound healing, clots, obesity, and diabetes [343]. Strawberry ameliorates peripheral insulin resistance, reduces α-amylase and α-glucosidase activity, and increases glucose-stimulated insulin release [343,344,345]. Quercetin, kaempferol, rutin, gallic acid, chlorogenic acid, caffeic acid, ellagitannins, and gallotannins found in strawberries may be responsible for the antioxidant, cardioprotective, antimetabolic syndrome, and neuroprotective properties of this plant [343,344,345,346,347].

45.
***Glycine max* (Soya bean)**


*Glycine max* (Fabaceae), also called soya bean, is employed to produce vegetable oils, tofu, soy milk, and soy sauce. It is also a remedy for osteoporosis, cardiovascular disease, and diabetes [348]. It contains a high content of proteins which improves diabetes and its complications by modulating various cell signaling pathways and regulating glucose homeostasis [263,349]. Soya beans are also able to mitigate obesity-induced metabolic disorders [350] as they lower triglyceride levels and have fatty acid synthase inhibitory activity, which contribute to ameliorating diabetes-related complications [351]. Among the soya bean proteins, β-conglycinin is the major constituent that has been reported to reduce insulin resistance and improve glucose uptake in skeletal muscles through AMPK activation [349].

46.
***Helianthus annuus* (Sunflower)**


*Helianthus annuus* (Asteraceae) is commonly known as sunflower. Sunflower seeds are often ingested to ameliorate diabetes, nephrotoxicity, cardiovascular disease, and hematologic disorders [352]. Sunflower is popular for its antitumor, antimicrobial, antioxidant, and anti-inflammatory effects. Sunflower seeds have been reported to lower body weight and body mass index (BMI) and have free radical scavenging activity. They can also reduce AGE formation and lower fasting blood glucose levels [353,354,355]. Sunflower is rich in flavonoids, alkaloids, saponins, tocopherols, carotenoids, tannins, chlorogenic acid, and caffeic acid. Tocopherols have been reported to improve insulin sensitivity and protect β-cells from oxidative stress [355].

47.
***Hibiscus rosa-sinensis Linn* (China rose)**


*Hibiscus rosa-sinensis Linn.*, also called China rose, China hibiscus, rose mallow, or shoe flower, belongs to the Malvaceae family. It is a popular traditional remedy for tumors, hair loss, infertility, diabetes, and wound healing [356,357,358]. It is reported to stimulate pancreatic-β cells, enhancing insulin secretion and glycogen accumulation in the liver. The antidiabetic properties of China rose may be attributed to its rich content of quercetin, cyanidin, ascorbic acid, gentisic acid, lauric acid, thiamine, niacin, margaric acid, calcium oxalate, and hentriacontane. Cyanidin, also present in China rose, has been demonstrated to improve endothelial function and oxidative damage [358,359,360].

48.
***Hylocereus undatus* (Dragon fruit)**


*Hylocereus undatus* (Cactaceae), also called dragon fruit or strawberry pear, is ethnomedicinally useful as a hypoglycemic, diuretic, antigastritis, wound healing and laxative agent [361,362]. It shows antidiabetic activity by regulating oxidative stress, reducing intestinal glucose absorption and plasma glucose levels, and improving insulin secretion. These effects can be attributed to several phytoconstituents, including phthalic acid, α-amyrin, oleic acid, linoleic acid, palmitic acid, gallic acid, syringic acid, *p*-coumaric acid, lycopene, β-carotene, and betacyanin [363].

49.
***Ipomoea batatas* (Sweet potato)**


*Ipomoea batatas* is a plant of the Convolvulaceae family, also known as sweet potato. This plant is a popular ethnomedicine for diabetes, diarrhea, splenosis, stomach distress, anemia, hypertension, and throat tumors [364,365]. Anthraquinones, coumarins, flavonoids (quercetin, lutein), saponins, tannins, phenolic acids, chlorogenic acid, terpenoids, β-carotene, zeaxanthin, and anthocyanins present in sweet potato may also substantially mitigate insulin resistance and regulate blood glucose levels by stimulating the production of insulin by pancreatic β-cells [366,367,368].

50.
***Juglans regia* L. (Walnut)**


The walnut plant or *Juglans regia* L. (Juglandaceae) is a reputed remedy for bacterial infection, stomachaches, thyroid disorders, diabetes, cancer, heart conditions, and sinusitis [369]. Its nut is high in fiber which makes it one of the best superfoods for diabetes control. One study reported that it improves glucose uptake, inhibits α-glucosidase, α-amylase, and protein tyrosine phosphatase 1B (PTP1B) activity, and reduces plasma glucose levels in streptozotocin-induced rats [370]. Gallic acid, caffeoylquinic acid, coumaroylquinic, juglone, and quercetin were identified as the potential bioactive compounds responsible for the antidiabetic, anti-inflammatory, and antioxidant effects of walnuts [371,372].

51.
***Lactuca sativa* (Lettuce)**


*Lactuca sativa* or lettuce is a leafy vegetable from the Asteraceae family, often served as a salad. The leaves and seeds of lettuce are used for treating hyperglycemia, osteodynia, and inflammatory conditions [373]. Lettuce inhibits the activity of α-amylase, α-glucosidase, and dipeptidyl peptidase-4 (DPP-4) enzymes. It can regulate postprandial glucose, fasting blood glucose, triglycerides, serum insulin, and cholesterol levels. These effects may be due to the presence of flavonoids such as quercetin, anthocyanins, and hydroxycinnamoyl derivatives [374,375,376,377].

52.
***Lagenaria siceraria* (Bottle gourd)**


*Lagenaria siceraria* (Cucurbitaceae) is popularly known as bottle gourd and regarded as a remedy for diabetes, jaundice, constipation, flatulence, insomnia, ulcer, piles, colitis, insanity, hypertension, congestive cardiac failure, skin diseases, and headaches [378,379]. Bottle gourd improves insulin production and glucose tolerance and suppresses intestinal glucose absorption. These effects may be attributed to isovitexin, isoorientin, saponarin, fucosterol, campesterol, cucurbitacin B, cucurbitacin D, cucurbitacin E, isoquercitrin, kaempferol, gallic acid, and protocatechuic acid [381,382].

53.
***Laurus nobili* (Bay leaves)**


*Laurus nobilis* or bay leaf is an important spice from the Lauraceae family. It is a popular aid for stomachaches, phlegm, colds, sore throats, headaches, indigestion, flatulence, eructation, epigastric bloating, and diabetes [383]. It is reported to decrease serum glucose levels, inhibit α-glucosidase, and stimulate the production of insulin by pancreatic β-cells. It is rich in phytoconstituents that include linalool, sabinene, kaempferol, quercetin, apigenin, luteolin, lauric acid, palmitic acid, linoleic acid, and the carotenoid lutein [384,385].

54.
***Litchi chinensis* (Lychee)**


*Litchi chinensis* (Sapindaceae), or lychee, is a seasonal fruit and useful ethnomedicine for coughs, ulcers, flatulence, testicular swelling, diabetes, hernias, and obesity [386]. Lychee seeds improve insulin resistance, glucose tolerance, and fasting blood glucose and serum triglyceride levels. Lychee has antihyperglycemic, antineurotoxic, anti-inflammatory, lipid-lowering, insulin-secreting, and α-glucosidase inhibitory properties. These effects may be attributed to the presence of flavonoids, triterpenes, sterols, and phenolic compounds [387,388].

55.
***Luffa acutangula* (Ridge gourd)**


*Luffa acutangula* (Cucurbitaceae), known as ridge gourd, is a valuable traditional medicine for diabetes, jaundice, hemorrhoids, urinary bladder stones, granular conjunctivitis, constipation, and leprosy. Ridge gourd has been reported to substantially lower serum glucose levels by enhancing insulin secretion and peripheral glucose uptake, as well as suppressing glycogenolysis and gluconeogenesis in alloxan-induced diabetic rats [389]. These effects may be attributed to the levels of apigenin, luteolin, myristic acid, α-pinene, carotene, oleanolic acid, β-myrcene, and linalool in its leaves, seeds, and fruit, which reduce blood glucose and oxidative stress [390].

56.
***Malus domestica* (Apple)**


The apple, *Malus domestica* (Rosaceae), is one of the most widely cultivated and commercially significant fruits. It is also a valuable folk medicine for wounds, diabetes, asthma, obesity, and cardiovascular disease [391,392,393]. Apple has been reported to significantly lower plasma glucose levels by increasing glucose-dependent insulinotropic polypeptide (GIP) and glucagon-like peptide-1 (GLP-1). Its antidiabetic effect has been linked with the flavonoid quercetin [394,395,396]. Apple also has antihypertensive, antioxidant, and anti-inflammatory properties, which may be attributed to several compounds, including quercetin, catechin, epicatechin, procyanidin, coumaric acid, chlorogenic acid, and gallic acid [394,395,396,397,398,399,400].

57.
***Mangifera indica* (Mango)**


*Mangifera indica* (Anacardiaceae), known as mango, is a delicious fruit and a plant used in folk medicine for asthma, dysentery, anthrax, indigestion, diarrhea, diabetes, and colic [401,402,403]. Mango pulps, stems, and peels improve postprandial glucose and insulin sensitivity in T2DM patients by inhibiting α-amylase and α-glucosidase [404,405,406]. Mango has been reported to exert antidiabetic activity by improving insulin secretion from clonal β-cells and isolated mouse islets, and by regulating fasting blood glucose, plasma insulin, liver glycogen levels, starch digestion, glucose absorption, body weight, and free radical scavenging activity in diabetic rats [405]. Another study in streptozotocin-induced diabetic rats reported its promising ability to decrease postprandial hyperglycemia [406]. The mentioned therapeutic effects of mango may be mediated by mangiferin, flavonoids, tannins, and alkaloids [405].

58.
***Mentha spicata* (Mint leaves)**


*Mentha spicata*, or mint, is a plant from the Lamiaceae family. It is known as a remedy for common colds, asthma, fever, obesity, digestive problems, dementia, hypertension, diabetes, and insomnia [407]. Mint boasts a range of health benefits. Mint leaves increase HDL cholesterol levels and reduce triglycerides, LDL, and VLDL cholesterol levels. It has antibacterial, antifungal, antioxidant, hepatoprotective, cytotoxic, anti-inflammatory, larvicidal, antigenotoxic, and antiandrogenic effects. Its ability to suppress α-amylase and α-glucosidase may be due to the presence of carvone, limonene, 1,8-cineole, pulegone, β-bourbonene, β-pinene, dihydrocarveol, and piperitone [408,409]. 

59.
***Moringa oleifera* Lam. (Moringa)**


*Moringa oleifera* Lam. (Moringaceae), also known as moringa or the drumstick tree, grows in many tropical and subtropical regions. It is regarded as a folk remedy for diabetes, liver disease, cancer, inflammation, hypercholesteremia, and hypertension [410,411]. Tannins, β-carotene, vitamin C, quercetin, and chlorogenic acid in moringa leaves aid diabetes through the inhibition of α-amylase and α-glucosidase enzymes. They also reduce serum glucose and fasting blood glucose levels [412,413,414].

60.
***Momordica charantia* (Bitter gourd)**


*Momordica charantia* or bitter gourd (Cucurbitaceae) has medicinal value for managing T2DM, dyslipidemia, cancer, obesity, malaria, dysentery, hypertension, and womb and worm infections [415,416,417,418]. Bitter gourd suppresses the intestinal absorption of glucose, inhibits gluconeogenesis, and reduces the accumulation of fats in adipocytes. It also activates the HMP and the PPARα pathways, regenerates pancreatic β-cells, and enhances glucose uptake in skeletal muscles. These effects may be attributed to the presence of phytoconstituents such as saponins, triterpenes, flavonoids, ascorbic acid, and steroids [419,420,421,422,423].

61.
***Morus alba* (Mulberry)**


*Morus alba* (Moraceae), also known as mulberry, is widely used as a remedy for diabetes, insomnia, tinnitus, dizziness, and for premature aging. It improves fasting blood glucose, total triglycerides, cholesterol, and HDL cholesterol levels via the IRS-2, GLUT4, and Akt pathways [424]. Quercetin and isoquercetrin present in mulberry leaves are reported to have insulin-releasing, antihyperlipidemic, antithrombotic, antiobesity, antioxidant, and anti-inflammatory effects, which may be beneficial in diabetic complications [425,426]. The bark of mulberry also lowers cholesterol and blood glucose levels, probably due to the presence of alkaloids, flavonoids, coumarins, anthocyanins, benzofurans, and phenolic acids [424,427].

62.
***Murraya koenigii* (Curry leaves)**


*Murraya koenigii* L. or the curry leaf plant belongs to the Rutaceae family. This plant is popular as an herbal remedy for piles, inflammation, itching, diabetes, and snake bites [428,429]. It has antimicrobial, antioxidant, antihyperglycemic, apoptotic, anticarcinogenic, anti-inflammatory, and antitumor effects. It has been also reported to protect against β-cell damage, enhance antioxidant defense systems, and reduce oxidative stress, as well as improving blood sugar levels in diabetic rats [430]. Bioactive substances such as mahanine, mahanimbine, murrayanol, koenigicine, quercetin, apigenin, kaempferol, catechin, and oliolide in curry leaves have been reported to synergistically regenerate β-cells, aid diabetic complications, and possess antihyperlipidemic effects [430,431].

63.
***Myristica fragrans* Houtt. (Nutmeg)**


*Myristica fragrans* Houtt. (Myristicaceae), known as nutmeg, is a flavoring spice and reputed folk remedy for skin infections, diarrhea, diabetes, Alzheimer’s disease, rheumatism, asthma, colds, coughs, and malaria [432]. Nutmeg demonstrates antidiabetic effects by enhancing insulin sensitivity, regulating blood glucose levels, and exhibiting antioxidant properties that protect against oxidative stress in diabetes. It strongly inhibits the release of pro-inflammatory cytokines such as IL-6 and TNF-α, and helps ameliorate β-cell function, inflammation, and obesity [433,434,435]. Nutmeg is a source of flavonoids, terpenes, phenylpropanoids, coumarins, lignans, alkanes, and indole alkaloids that can elicit antiprotozoal, antimicrobial, immunomodulatory, anxiolytic, and neuroprotective effects [432].

64.
***Nigella sativa* L. (Black seeds)**


*Nigella sativa* L. (Ranunculaceae) or black seeds are a reputed herbal remedy for asthma, dyslipidemia, diabetes, and diarrhea [436]. Black seeds exert antidiabetic effects by reducing carbohydrate digestion and absorption in the gut, improving insulin secretion, and enhancing glucose tolerance in T2DM animal models. Other antidiabetic effects of black seeds include lowering lipid and blood glucose levels, suppressing hepatic gluconeogenesis, and inhibiting α-amylase and α-glucosidase, as well as boosting insulin production and sensitivity. These effects can be attributable to phytochemicals that include thymoquinone, thymol, limonene, carvacrol, *p*-cymene, longifolene, α-pinene, linoleic acid, oleic acid, palmitic acid, saponins, and alkaloids. Thymoquinone in black seeds is known to enhance insulin secretion and insulin sensitivity through activating the PI3K/Akt signaling pathway [437,438,439,440].

65.
***Ocimum sanctum* L. (Holy basil)**


*Ocimum sanctum* L., known as holy basil or Tulsi, belongs to the Lamiaceae family. Tulsi is traditionally used for anxiety, coughs, asthma, diarrhea, fevers, dysentery, arthritis, eye diseases, indigestion, back pain, skin disorders, ringworm, insect, snake, and scorpion bites, malaria, vomiting, gastritis, diabetes, and cardiac and genitourinary infections [441,442]. Tulsi leaves help improve insulin synthesis and pancreatic β-cell activity, as well as inhibiting intestinal glucose absorption. Its phytoconstituents such as eugenol, ursolic acid, carvacrol, linalool, caryophyllene, triterpenoids, and tannins may contribute to these effects [443,444].

66.
***Olea europaea* L. (Olive)**


*Olea europaea* L. (Oleaceae), or olive, is traditionally used to treat diabetes, diarrhea, inflammation, urinary tract infection, hypertension intestinal diseases, hemorrhoids, and rheumatisms [445,446,447]. It offers a promising range of health benefits such as anti-inflammatory, antidiabetic, and immunomodulatory properties [448,449,450]. Olive oil notably prevents hepatic gluconeogenesis and inhibits glucose-6-phosphatase activity. It enhances catalase activity and regulates body weight and plasma glucose levels, possibly due to the presence of oleanolic acid, cinnamic acid, and secoiridoid glycosides such as oleuropein [448,449,450].

67.
***Origanum vulgare* (Oregano)**


*Origanum vulgare* (Lamiaceae), known as oregano, is a folk medicine for acne, cystic fibrosis, diabetes, and bacterial infections [442,451]. It alleviates diabetic complications, including nephropathy, atherosclerosis, and retinopathy, by inhibiting α-glucosidase, thereby reducing the breakdown of complex carbohydrates into glucose, and lowering both glycosylation and oxidative stress. Moreover, it improves glucose uptake in skeletal muscles by increasing GLUT2 levels, leading to better control of blood sugar levels [453]. Oregano is a source of amburoside A, apigenin, luteolin 7-*O*-glucuronide, rosmarinic acid, and lithospheric acid, which have antimicrobial, antifungal, antioxidant, anti-inflammatory, and antiviral properties [454,455].

68.
***Passiflora edulis* (Passion fruit)**


*Passiflora edulis* (Passifloraceae), commonly known as passion fruit, is used as an ethnomedicine for coughs, diabetes, dysmenorrhea, dysentery, arthralgia, and constipation [456,457]. Previous studies have shown that it reduces weight gain and lipid accumulation, as well as improving insulin sensitivity and glucose tolerance via the Sirt1 and p-AMPK pathways [456,458]. It contains more than 110 bioactive constituents, including piceatannol, tocopherols, β-carotene and other carotenoids, gallic acid, flavonoids such as rutin and quercetin, and coumaric acid, which have antidiabetic, antioxidant, antihypertensive, antimicrobial, hepatoprotective, and lung-protective qualities [457,459,460,461,462,463]. A reduction in blood glucose levels has been linked to the presence of piceatannol, which is present in high amounts in passion fruit [457].

69.
***Persea americana* (Avocado)**


*Persea americana* (Lauraceae) or avocado is a popular fruit and a remedy traditionally used to manage cardiovascular diseases and diabetes [464]. Avocado has been reported to lower blood glucose levels and regulate glucose uptake in the liver and skeletal muscles, as well as restoring intracellular energy homeostasis through activation of the PKB/Akt pathway [470]. Histopathological analysis of diabetic rats also revealed regeneration of clonal pancreatic β-cells following avocado treatment. Avocado seed, bark, and leaf extracts contain flavonoids, alkaloids, saponins, tannins, and glycosides, which are known for their antihyperglycemic properties [466,467,468].

70.
***Petroselinum crispum* (Parsley)**


*Petroselinum crispum* (parsley) is a plant from the Apiaceae family. As well as being a culinary herb, it is an ethnomedicine traditionally used for diabetes, urinary tract infections, dysmenorrhea, hypertension, dermatitis, and gastrointestinal disorders [469]. Parsley exerts long-lasting control of sugar levels by regulating plasma glucose, body weight, and electrolyte (sodium and potassium) balance. It also promotes glucose uptake in muscles by inhibiting gluconeogenesis (sugar production) and stimulating glycolysis (sugar breakdown) [470,471]. The main bioactive constituents of parsley are coumarins, phthalides, phenylpropanoids, and tocopherols, with antimicrobial, antihepatotoxic, antihypertensive, antihyperlipidemic, hypouricemic, and antioxidative properties [472].

71.
***Phaseolus vulgaris* L. (Kidney bean)**


*Phaseolus vulgaris* L. (Fabaceae), or the kidney bean, is another nutritious legume crop, which is ethnomedicinally used for wounds, pharyngitis, fevers, obesity, diabetes, cancer, and vaginal infections [473,474]. Beyond their potential to lower blood sugar levels, kidney beans exhibit a range of other health benefits, including anti-obesity and anti-inflammatory properties [473,474,475,476,477,478,479]. They are a potential source of protocatechuic acid, *p*-coumaric acid, procyanidin, myricetin, naringenin, gallic acid, quercetin, catechin, kaempferol, and ferulic acid, which may contribute to alleviating diabetic complications via inhibiting α-glucosidase, enhancing insulin sensitivity in peripheral tissues, delaying the absorption of glucose, and reducing gluconeogenesis [475,476].

72.
***Phoenix dactylifera* (Date)**


*Phoenix dactylifera* or date palm is a flowering plant belonging to the Arecaceae family. Date palm is a traditional medicine for fever, inflammation, nervous disorders, and dementia [477]. In vitro studies demonstrated that date fruit has α-glucosidase and α-amylase inhibitory activity, reduces the intestinal absorption of glucose, and improves pancreatic β-cell function, insulin secretion, and β-cell numbers [478,479]. The antihyperglycaemic, antioxidant, anti-inflammatory, hepatoprotective, and nephroprotective properties of date palm may be attributable to its vast array of phytochemicals, including oleic acid, linoleic acid, catechin, epicatechin, anthocyanin, ellagic acid, gallic acid, *p*-coumaric acid, coumarins, quercetin, rutin, myricetin, apigenin, naringenin, and chlorogenic acid [477,480].

73.
***Phyllanthus emblica* L. (Amla)**


*Phyllanthus emblica* L. (Phyllanthaceae), commonly called Indian gooseberry or amla, is a remedy for coughs, peptic ulcers, skin diseases, jaundice, diarrhea, dysentery, diabetes, cardiac disorders, and premature aging [481,482]. Recent studies suggest that the fruit, bark, leaves, and roots of amla significantly reduce plasma glucose levels through the inhibition of α-amylase and α-glucosidase activity and activation of the AMPK signaling pathway. The main phytoconstituents in amla, such as gallic acid, ellagic acid, pectin, quercetin, linoleic, oleic acid, and myristic acid, are effective in reducing inflammation and blood glucose levels and increasing insulin sensitivity [483,484].

74.
***Piper betle* L. (Betel leaf)**


*Piper betle* L. (Piperaceae), also known as betel leaf, is widely used as a folk medicine for wounds, bronchitis, diabetes, coughs, indigestion in children, headaches, arthritis, and joint pain [485]. It increases insulin production, improves glucose tolerance, and decreases blood glucose levels substantially [486]. Betel leaf contains many phytoconstituents such as eugenol, selinene, hydroxychavicol, cadinene, caryophyllene, estragole, linalool and other terpenes, phenols, steroids, saponins, and tannins, which may play an important role in the management of diabetic complications [487,488].

75.
***Pisum sativum* L. (Pea)**


*Pisum sativum* L., known as the pea, is a plant that belongs to the Fabaceae family. Peas are a reputed remedy for diabetes, gastrointestinal disorders, hyperlipidemia, and blood diseases [489]. Phytoconstituents such as quercetin, ellagic acid, coumaric acid, β-sitosterol, β-amyrin, catechin, myricetin, vanillic acid, and kaempferol may be responsible for the antidiabetic properties of peas. It remarkably improves plasma glucose levels, glucose tolerance, glucose uptake, glucose homeostasis, and diabetic complications [490,491]. It is also known to alleviate weight loss, polyphagia, triglycerides, and LDL cholesterol levels via interacting with AMPK, α-glucosidase, IRS-1, and IRS-2 [492].

76.
***Prunus armeniaca* L. (Apricot)**


*Prunus armeniaca* L. (Rosaceae), known as apricot, is a promising antidiabetic, cardioprotective, hepatoprotective, nephroprotective, antioxidant, antimicrobial, anti-inflammatory, anticancer, and antiviral remedy [493,494]. Apricot has been reported to stimulate insulin secretion, reduce oxidative stress, and show α-glucosidase inhibitory activity in alloxan-induced diabetic mice. It is rich in coumaric acid, benzyl glycosides, cyanogenic glycosides, vanillin, catechin, epicatechin, neochlorogenic acid, chlorogenic acid, rutin, quercetin, and lutein [494,495].

77.
***Prunus domestica* (Plum)**


*Prunus domestica* (Rosaceae), or plum, is a fruit and a beneficial ethnomedicine for anemia, Alzheimer’s disease, irregular menstruation, diabetes, and constipation [496,497,498]. Recent studies reported that plum reduces oxidative stress and inhibits α-glucosidase, α-amylase, pancreatic lipase, and HMG-CoA reductase, lowering LDL, cholesterol, and triglyceride levels [499,500]. Catechin, epicatechin, chlorogenic acid, kaempferol, quercetin, and β-carotene present in plum may contribute to its antihyperglycemic, anti-inflammatory, antioxidant, and lipid-lowering properties [501,502,503].

78.
***Prunus dulcis* (Almond)**


*Prunus dulcis*, or almond, is a plant from the Rosaceae family that is used as a remedy for neurological and respiratory disorders, diabetes, and urinary tract infections [504]. Almond has a high fiber content, which helps in ameliorating diabetes by suppressing appetite and lowers blood sugar levels via increasing insulin production and decreasing the stomach’s emptying time. Its pharmacological effects include antioxidant, anti-inflammatory, hepatoprotective, anxiolytic, and nerve-improving properties. Almonds are rich in oleic acid, linoleic acid, *p*-coumaric acid, anthocyanins, kaempferol, quercetin, and chlorogenic acid [504,505].

79.
***Prunus persica* (Peach)**


*Prunus persica* or peach is a species from the Rosaceae family that is very useful in improving blood circulation, blood clotting, constipation, and diabetes [506]. Peach inhibits α-glucosidase and α-amylase activity and enhances insulin production by increasing the regeneration of pancreatic islet β-cells [507,508]. Various bioactive compounds in peaches, such as procyanidins, epicatechin, catechin, chlorogenic acid, quercetin, and kaempferol, play a vital role in the secretion of insulin from clonal pancreatic β-cells and have demonstrated DPP-IV inhibitory activity [507,509].

80.
***Punica granatum* (Pomegranate)**


*Punica granatum* or pomegranate (Lythraceae) is traditionally used for dysentery, diarrhea, piles, bronchitis, biliousness, and diabetes [510,511]. Recent studies have shown that it can stimulate insulin secretion, enhance glucose transporter type 4 (GLUT-4) translocation, and regulate blood glucose levels. The phytoconstituents isolated from pomegranate, such as ellagic acid, gallotannins, anthocyanins, quercetin, kaempferol, luteolin glycosides, linolenic, arachidic, and palmitoleic acids, may contribute to the insulin-releasing and glucose-lowering properties of this plant [512,513].

81.
***Psidium guajava* (Guava)**


*Psidium guajava* (Myrtaceae), commonly known as guava, is widely used for dysentery, diabetes, and diarrhea [514,515,516]. Studies conducted on its leaves have revealed that it activates the AMPK and PI3K/AKT signaling pathways, improves hepatic glycogen accumulation, and regulates the activity of superoxide dismutase (SOD), glucose transporter 2 (GLUT-2), and fasting blood sugar levels [517,518,519,520]. The antidiabetic activity of guava may be attributed to compounds such as quercetin, avicularin, guaijaverin, tannins, and triterpenes [521,522].

82.
***Raphanus sativus* L. (Radish)**


*Raphanus sativus* L. (Brassicaceae), also called radish, has been employed as an effective remedy for diabetes, jaundice, gastric disorders, dyspepsia, and liver enlargement since ancient times [523]. Radish seeds significantly decrease hyperglycemia by reducing insulin resistance, limiting intestinal glucose absorption, and increasing glucose uptake in skeletal muscles [524]. Myricetin, catechin, epicatechin, quercetin, *p*-coumaric acid, β-carotene, camphene, anthocyanin, glucosinolates, and isothiocyanate are some of the phytoconstituents in radish which that have been demonstrated to possess antioxidant, anti-inflammatory, and radical-scavenging activity [525,526].

83.
***Rosmarinus officinalis* L. (Rosemary)**


*Rosmarinus officinalis* L., familiar as rosemary, is an important herb from the Lamiaceae family and is commonly recognized as a flavor enhancer, food preservative, wound healer, and antihyperglycemic and analgesic agent. It is also efficacious against mycosis, alopecia, ultraviolet damage, skin cancer, inflammatory diseases, and diabetes [527,528]. Rosemary has been suggested to act via several pathways to improve blood sugar control. It reduces Irs1 protein levels, which can contribute to insulin resistance. It also recruits GLUT-4 receptors to the surface of muscle cells, facilitating glucose uptake from the bloodstream. Additionally, it activates pathways (pAKT and pAMPK) that promote glucose uptake and inhibit gluconeogenesis. These overall effects improve glucose utilization, leading to lower blood sugar levels [529,530,531]. Moreover, rosemary contains several types of flavonoids, carnosol, and carnosoic, rosmarinic, ursolic, oleanolic, and micromeric acids. The presence of bioactive compounds may be responsible for its antimicrobial, antitumor, antithrombotic, antidepressant, and antioxidant effects [527,532].

84.
***Rubus fruticosus* (Blackberry)**


*Rubus fruticosus* or blackberry is a member of the Rosaceae family and well-known for its use in mouthwash to relieve gum inflammation and mouth ulcers. It is also used for sore throats, respiratory disorders, anemia, diarrhea, dysentery, cystitis, diabetes, and hemorrhoids [533]. Blackberry has α-amylase, α-glucosidase, and β-glucosidase inhibitory activity, and reduces oxidative stress. This has been associated with its high content of anthocyanins, cyanidins, kaempferol, quercetin, myricetin, *p*-coumaric acid, rutin, and gallic acid [534,535,536].

85.
***Salvia hispanica* L. (Chia seeds)**


*Salvia hispanica* L. (Lamiaceae), also known as chia seeds, has a high nutritional and medicinal value. Chia seeds are used to treat indigestion, hyperlipidemia, and diabetes [537,538]. Chia seeds decrease fasting plasma glucose and LDL cholesterol levels, inhibit the production of pro-inflammatory cytokines (e.g., IL-6, Interleukin-2, and TNF-α), reduce body weight, and have α-amylase and α-glucosidase inhibitory activity [539,540]. They are a source of myricetin, quercetin, kaempferol, chlorogenic acid, and caffeic acid, which have hepatoprotective, antidiabetic, antihypertensive, and antioxidant effects. They also contain omega-3 fatty acids, which can enhance insulin sensitivity and reduce inflammation [538].

86.
***Sesamum indicum* (White sesame seeds)**


Sesame seeds, also called *Sesamum indicum* (Pedaliaceae), are traditionally used for wounds, amenorrhea, ulcers, asthma, hemorrhoids, inflammation, and diabetes [541,542]. Sesamin, the main bioactive compound in sesame seeds, can significantly ameliorate diabetes by enhancing insulin sensitivity, reducing inflammation, boosting antioxidant defenses, and regulating lipid metabolism [543]. Other phytochemicals in sesame seeds include other lignans, such as sesamolin, and phytosterols. These are reported to decrease fasting and postprandial blood glucose, reduce cholesterol and oxidative stress, and improve renal disorders, fat metabolism, cell viability, and insulin secretion [544,545,546].

87.
**
*Solanum lycopersicum L. (Tomato)*
**


*Solanum lycopersicum* L. (Solanaceae) or tomato is vastly produced for consumption worldwide and is also a beneficial remedy for dermatitis, cancer, hypertension, and hyperglycemia [547,548,549]. The underlying mechanisms of its hypoglycemic effects are through regulation of the PI3K/Akt, FOXO1, and PPAR-γ signaling pathways. Tomato enhances insulin signaling, improves glucose uptake, and modulates lipid metabolism [549]. Due to its high lycopene content, tomato may help mitigate diabetes-induced inflammation. Additionally, the presence of carotenoids may also contribute to improving insulin sensitivity [550,551]. Tomato also contains ferulic acid, β-carotene, tomatine, kaempferol, quercetin, naringenin, *p*-coumaric acid, and caffeic acid, which exert antioxidant, anti-inflammatory, antihyperglycemic, and neuroprotective effects [552,553].

88.
***Solanum melongena* (Eggplant)**


*Solanum melongena* (Solanaceae) or eggplant is a nutritious vegetable and an efficient remedy for arthritis, diabetes, dyslipidemia, bronchitis, and asthma [554]. It has been reported to inhibit α-amylase and α-glucosidase enzymes, inhibit gluconeogenesis, increase the translocation of GLUT4, increase glucose uptake in skeletal muscle, and reduce fatty acids, triglycerides, and cholesterol levels [555]. The bioactive constituents present in eggplant include thiamin, niacin, chlorogenic acid, saponins, solasodine, and delphinidin. These constituents have been associated with anti-inflammatory, antioxidant, antihypertensive, antihyperlipidemic, anti-obesity, and hepatoprotective effects [556,557,558].

89.
***Spinacia oleracea* (Spinach)**


*Spinacia oleracea* (spinach) belongs to the Chenopodiaceae family. It is a folk remedy for bloody stools, diarrhea, stomachaches, obesity, and diabetes [559]. It notably improves diabetic retinopathy and hyperglycemia by modulating multiple pathways such as inhibition of excess AGE and carbonyl group production, glycation, and thiol group depletion in bovine serum albumin [560]. Spinach aids insulin resistance by inhibiting increased levels of serum C-reactive protein, tumor necrosis factor (TNF)-α, and Interleukin-6 [561]. Moreover, it is rich in β-carotenoids, lutein, zeaxanthin, vitamins, and minerals that also exert hypoglycemic, hypolipidemic, anti-obesity, and antioxidant effects [562,563,564].

90.
***Syzygium aromaticum* (Clove)**


*Syzygium aromaticum* flower buds (Myrtaceae), typically known as clove, are a seasoning spice and an efficacious aid for increased gastritis, diabetes, and indigestion [565]. Clove is reported to improve insulin sensitivity, inhibit aldose reductase, prevent diabetic complications such as neuropathy and nephropathy, regulate SIRT1 to enhance glucose metabolism, and promote muscle glucose uptake, all of which assist the management of diabetes. Phytoconstituents in clove include alkaloids, terpenes, tannins, phenolics, steroids, flavonoids, glycosides, and saponins, which may mitigate diabetic complications by decreasing insulin resistance [565,566,567,568]. Among them, eugenol acetate, eugenol, and gallic acid act via PPAR-γ activation, aldose reductase inhibition, sirtuin 1 (SIRT1) regulation, and muscle glycolysis [566,567,568].

91.
***Syzygium cumini* (Java plum)**


*Syzygium cumini* (Java plum) belongs to the Myrtaceae family and is used to treat asthma, bronchitis, sore throats, biliousness, dysentery, diabetes, and ulcers [569]. Its pharmacological actions, such as stimulating clonal pancreatic β-cells to release insulin, have been compared to those of sulfonylureas and biguanides [570]. A recent study reported that Java plum seeds are effective in reducing plasma and urine glucose levels in diabetic rabbits [571]. The Java plum is a good source of phytoconstituents such as anthocyanins, malvidin-3-glucoside, petunidin-3-glucoside, ellagic acid, and the flavonoids isoquercetin, kaempferol, and myricetin, which may be responsible for its antioxidant, antibacterial, gastroprotective, and antidiarrheal properties [569].

92.
***Tamarindus indica* L. (Tamarind)**


*Tamarindus indica* L., also known as tamarind, belongs to the Fabaceae family. This plant is mostly cultivated in the Indian sub-continent and other tropical regions. It is known to effectively treat inflammation, stomach pain, sore throats, rheumatism, wounds, diarrhea, dysentery, fever, malaria, respiratory conditions, constipation, and eye diseases [572]. Beyond its culinary uses, tamarind offers a range of health benefits due to its antioxidant and anti-inflammatory properties that aid digestion and the expulsion of mucus [577,578,579,580]. The presence of apigenin, anthocyanin, procyanidin, catechin, epicatechin, taxifolin, eriodyctiol, and naringenin help to control DM by inhibiting the activity of α-amylase and α-glucosidase [573,574,575]. Among them, catechin, anthocyanin, and epicatechin notably lower blood glucose levels via glucose-6-phosphatase inhibitory activity, improving blood glucose tolerance and promoting the regeneration of β-cells [577].

93.
***Theobroma cacao* (Cocoa)**


*Theobroma cacao* (Malvaceae) is typically known as cocoa beans and is commercially processed to make chocolate, particularly dark chocolate. It is a reputed remedy for measles, malaria, toothache, and diabetes. Its antidiabetic effect is via improving insulin secretion, GLUT4 translocation, and glucose uptake [577,578]. Moreover, it exerts inhibitory activity on α-amylase and α-glucosidase, reduces ROS generation, and increases GSH and Nrf2, thereby enhancing insulin secretion and β-cell survival [579,580]. Flavonoids, procyanidins, catechin, and epicatechin have been implicated in mitigating diabetic complications and have demonstrated antioxidant, anti-inflammatory, and hepatoprotective effects [581,582].

94.
***Trichosanthes cucumerina* L. (Snake gourd)**


*Trichosanthes cucumerina* L. (Cucurbitaceae) or snake gourd is an ethnomedicine for diabetes, bronchitis, headache, cathartic, anthelmintic, indigestion, ulcers, and stomach and skin disorders [583,584]. The roots, fruit, seeds, and leaf juice of snake gourd stimulate β-cell insulin secretion, enhance glucose uptake in peripheral tissues, and reduce intestinal glucose absorption. This antihyperglycemic effect may be attributed to its rich content of carotenoids, gallic acid, neochlorogenic acid, caffeic acid, *p*-coumaric acid, rutin, kaempferol, quercetin, ursolic acid, and oleanolic acid [583,584,585].

95.
***Trigonella foenum-graecum* (Fenugreek seeds)**


*Trigonella foenum-graecum* (Fabaceae), or fenugreek seeds, is reputed to be an effective tonic for ulcers, sinusitis, hayfever, diarrhea, diabetes, and kidney diseases [586]. Studies have documented their antidiabetic activity, with promising reductions in fasting and postprandial blood glucose, an enhancement in glucose uptake, glucose tolerance, and peripheral insulin action [592,593]. Phytoconstituents in fenugreek seeds, such as steroids, alkaloids, flavonoids, polyphenols, and saponins, have anti-obesity, antihyperlipidemic, antioxidant, anticancer, anti-inflammatory, and antifungal properties. Specific phytochemicals, including trigonelline, diosgenin, and galactomannan, have been shown to enhance insulin sensitivity, improve glucose metabolism, and reduce blood sugar levels [586,587,588,589].

96.
***Vaccinium corymbosum* (Blueberry)**


*Vaccinium corymbosum* (Ericaceae), also called blueberry, is a widely used fruit with medicinal properties that are useful for colds, inflammation, cardiovascular diseases, diabetes, and ocular disorders [590,591]. It exerts its antidiabetic activity by inhibiting α-amylase and α-glucosidase activity and ameliorating diabetic retinopathy [596,597]. It is rich in pectin, anthocyanins, anthocyanidins, protocatechuic acid, and petunidin, which may contribute to its antidiabetic, antiobesity, antioxidant, cardioprotective, neuroprotective, and immunomodulatory effects [592].

97.
***Vigna radiata* (Mung bean)**


*Vigna radiata* (Leguminosae), or mung bean, is an important legume crop with a high nutrient value. It is a helpful remedy for heatstroke, gastrointestinal disorders, dermatitis, hyperglycemia, hypertension, hyperlipidemia, and melanogenesis [593,594]. Mung bean significantly reduces serum glucose, total cholesterol, and triglycerides levels. It also inhibits gluconeogenesis and glycolysis, as well as α-glucosidase and α-amylase activity [595,596,597]. Mung bean is a rich source of proteins, vitamins, minerals, and bioactive compounds that include quercetin, myricetin, kaempferol, catechin, vitexin, isovitexin, coumaric acid, luteolin, and caffeic and gallic acid, which all help enhance insulin sensitivity and reduce oxidative stress and blood glucose levels [598,599].

98.
***Vitis vinifera* (Grapes)**


*Vitis vinifera* (Vitaceae), commonly called grapes, can aid in diarrhea, wounds, hepatitis, stomachaches, cardiovascular diseases, varicose veins, hemorrhoids, atherosclerosis, and diabetes [600]. It is known for regenerating clonal pancreatic β-cells and regulating plasma glucose levels by inhibiting the intestinal absorption of glucose [601]. The phytomolecules found in grapes, such as triterpenoid acids, gallic acid, catechin, epicatechin, gallocatechin, *p*-coumaric, and ferulic, acids may contribute to its anti-inflammatory, antioxidant, anticholesterolemic, and glucose-lowering properties [602].

99.
***Zea mays* (Corn)**


*Zea mays* (Poaceae), or corn, is a popular ethnomedicine for malaria, bladder stones, heart diseases, and diabetes [603,604]. Corn is a superfood that is rich in fiber and nutrients. Recent findings reveal that corn silk (the extended stigmas of *Z. mays* flowers) improves insulin resistance by lowering LDL cholesterol, total cholesterol, triglyceride, and malondialdehyde levels. It also reduces body weight and the accumulation of lipids in the liver [605]. Moreover, corn possesses antioxidant, anti-inflammatory, antimutagenic, anti-angiogenesis, and anticarcinogenic properties. One in vivo study revealed that the flavonoid glycoside hirsutrin was the main constituent beneficial in diabetic complications through suppressing aldose reductase and the formation of galactitol [606]. The antidiabetic properties of corn have been attributed to flavonoids, alkaloids, saponins, phenols, tannins, and phytosterols that could inhibit α-amylase and α-glucosidase and aid diabetic nephropathy [607,608].

100.
***Zingiber officinale* (Ginger)**


*Zingiber officinale* (Zingiberaceae), commonly called ginger, is a traditional treatment for muscular aches, arthritis, rheumatism, diabetes, hypertension, infections, and helminthiasis [609]. Ginger plays a significant role in regulating blood sugar levels by promoting the actions of GLUT-4 and PPAR-γ, which help muscles absorb glucose more efficiently. It also protects insulin-producing β-cells in the pancreas [610]. Ginger is rich in various compounds (e.g., gingerols) that have a range of pharmacological effects, including anti-inflammatory and neuro- and cardio-protective properties [611].

**Table 1 nutrients-16-03709-t001:** Traditional uses, pharmacological actions, and phytoconstituents of dietary plants.

Dietary Plants		Plant Parts Used	Traditional Uses	Pharmacological Actions	Diabetic Model	Treatment Dose	Duration of Treatment	Phytochemicals	References
Scientific Name	Common Name								
1. *Abelmoschus esculentus* L.	Okra	Fruit, roots	Chronic kidney disease, T2DM, cardiovascular diseases	Blood glucose↓, TC↓, TG↓, LDL-C↓, VLDL↓, HDL↑, body weight↓, α-amylase and α-glucosidase activity↓	STZ-induced T2DM mice (*n* = 10)	200–400 mg/kg/day	56 days	Oxalic acid, iodine, pectin, flavonoids, saponins, alkaloidsd-galactose, l-rhamnose, d-galacturonic	[142,143,144,612]
2. *Actinidia chinensis*	Kiwi	Fruit	Dyspepsia, vomiting, loss of appetite, diabetes	serum microRNA-424↑, Keap1↑, Nrf2↑, IL-6↓, IL-1↓, SOD↑, GSH↑, ALT↓, AST↓, inflammation ↓	T2DM patients (50–70 years old, *n* = 55–61)	10 mg/kg/day	270 days	Triterpenoids, polyphenols, β-carotene, lutein, xanthophylls, amino acids	[145,146,147,148]
3. *Aegle marmelos* L.	Stone apple	Fruit	Inflammation, asthma, hyperglycemia, febrifuge, hepatitis, analgesic, antifungal agent, colitis, flatulence, dysentery, fever	Glucose tolerance↑, α-amylase and α-glucosidase activities↓, insulin secretion↑, intestinal glucose absorption↓, BMI↓, polydipsia↓, polyphagia↓	STZ-induced T2DM diabetic rats (*n* = 9–11)	250–500 mg/kg/day	28 days	Marmelosin, psoralen, limonene, citronellal, citral, marmin, skimmianine, aegelin, fagarine, lupeol, cineol, halfordiol, citronellal, cuminaldehyde, eugenol, marmesinin	[149,150,151]
4. *Agaricus bisporus*	Mushroom	Rhizome	Cold, cough, influenza, asthma, cancer, diabetes, hepatic disorders	Blood glucose↓, TC↓, TG↓, LDL-C↓, insulin secretion↑, glucagon secretion↓	STZ-induced Sprague-Dawley rats (*n* = 6–8)	200 mg/kg/day	21 days	Lectins, β-glucans, polyphenols, *p*-hydroxybenzoic acid, protocatechuic acid, gallic acid, cinnamic, *p*-coumaric acid, ferulic acid, chlorogenic acid, and catechin	[152,153,154,155,156]
5. *Allium cepa*	Onion	Fruit	Wound healing, scars, keloids, bee sting inflammation, dysmenorrhea, vertigo, fainting, migraine, bruises, earache, jaundice, pimples, diabetes	Blood glucose↓, FBG↓, TC↓, TG↓ α-amylase and α-glucosidase activity↓, insulin secretion↑, β-cell protection↑, oxidative stress↓	Alloxan- induced diabetic rats (*n*= 27)	200–300 mg/kg/day	42 days	Quercetin, lectin, steroids, catechol, thiocyanate, isoflavones, humulone, quercetin, apigenin, rutin, myricetin, kaempferol, catechin, resveratrol, ajoene, phenolics, phenolic acids, and anthocyanins	[157,158,159,160,161]
6. *Allium sativum* L.	Garlic	Fruit	Cold, fever, headache, abdominal pain, sinus congestion, gout, rheumatism, hemorrhoids, asthma, bronchitis, cancers, cough cardiovascular diseases, arthritis, tuberculosis, rhinitis, malaria, dermatitis, enlarged spleen, fistula, UTI, kidney stone	Blood glucose↓, TC↓, TG↓, GLUT-4 activity↑, β-cell function↑, glucose uptake↑, creatinine↓, uric acid↓, urea↓, AST and ALT↓, insulin sensitivity↑, insulin secretion ↑, insulin production ↑, glucose tolerance↑,	STZ-induced Wistar rats (*n* = 6)	100–500 mg/kg/day	14 days	AJoene, cysteine, allicin, β-resorcylic acid, gallic acid, rutin, protocatechuic acid, quercetin	[162,163,164,165,166,167]
7. *Aloe barbadensis Mill.*	Aloe vera	Leaves	Wound healing, constipation, colic, worm infestation, dermatitis, hypertension	FBG ↓, TG↓, TC↓, AGE formation ↓, body weight, diabetic nephropathy↓	STZ-induced Wistar rats (*n* = 7)	300 mg/kg/day	49 days	Flavonoids, acemannam, flavones, quinone, galactan, pectin, ornanic acids	[166,168,169,170,171,172,173,613]
8. *Anacardium occidentale* L.	Cashew nut	Nut, leaves, bark	Fevers, aches, pains, diarrhea, diabetes, skin irritations, arthritis	Blood glucose↓, SOD↑, IR↓, gluconeogenesis↓, insulin secretion ↑	Alloxan-induced Wistar rats (*n* = 6)	100–250 mg/kg/day	40 h	Arginine, isoleucine, leucine, lysine, arachidic acid, lignoceric acid, gadoleic acid, linolenic acid, cyanidin, peonidin, anacardic acid, cardanol, limonene, lactone, palmitic acid	[174,175,176,177]
9. *Ananas comosus* L.	Pineapple	Fruit, peel, leaves	Pain, skin diseases, edema, wound, indigestion, diabetes and blood clotting	IR↓, insulin sensitivity↑, HDL-c↑, HbA1c↓, body weight↓, LPL activity↑, HMGCoA reductase activity↓	Alloxan- induced Wistar rats (*n* = 6)	400 mg/kg/day	15 days	Bromelain, flavonoids, coumaric acid, ellagic acid, ferulic acid, chlorogenic acid	[178,179,180,181,182,183,184]
10. *Apium graveolens*	Celery	Leaves, seeds, roots	Arthritis, spleen dysfunction, diabetes, sleep disturbances, CNS disorders	Blood glucose↓, PPBG↓, plasma insulin↑, GLUT-4 transloaction↑, mitochondrial dysfunction↓, insulin sensitivity↑, inflammation↓	Elderly diabetic patients above 60 years (*n* = 8, 5 female, 3 male)	250 mg/kg/3 times a day	12 days	Quercetin, thymoquinone, frocoumarin coumaric acid, gallic acid, flavonoids, alkaloids, steroids, limonene, selinene, glycosides	[185,186,187,188,189,190,614]
11. *Artocarpus heterophyllus*	Jackfruit	Fruit, leaves, bark, seeds, roots	Wound healing, cancer, diabetes	PPBG↓, FBG↓, IR↓, HbA1c↓, α-amylase and α-glucosidase activities↓, HDL-c↑, LDL↓	T2DM patients (18–60 years, *n* = 20)	30,000 mg/kg/day	84 days	Carotenoids, tannins, volatile acids, sterols, chrysin, silymarin, isoquercetin	[193,194,195,196,197]
12. *Asparagus officinalis*	Asparagus	Stem	Asthma, liver, rheumatic, kidney, bladder diseases	Blood glucose↓, β-cell function↓, FBG↓, TG↓, serum insulin↑, body weight↓, hepatic glycogen↓	STZ-induced Wistar rats (*n* = 6)	250–500 mg/kg/day	28 days	Asparagine, tyrosine, arginine, flavonoid, saponin, resin, tannin	[198,199,200,201,202]
13. *Avena sativa*	Oats	Grains	Dermatitis, cancer, diabetes, cardiovascular disease	PPBG↓, HbA1c↓, body weight↓, HDL↑, MDA↓, FBG↓, IR↓, TC↓, TG↓, LDquinol-C↓, SOD↑	T2DM patients (50–70 years, *n* = 14)	1 IU/kg/ day	28 days	β-glucan, tocopherols, tocotrienols, phenolic acids, sterols, selenium, avenanthramides	[203,204,205,206,207,615]
14. *Averrhoa carambola* L.	Star fruit	Fruit	Chronic headache, fever, cough, gastroenteritis, diarrhea, diabetes, ringworm infections, skin inflammations hypertension, hyperglycemia	Blood glucose↓, TG↓, TC↓, FFAs↓, serum insulin↑, glucose uptake↑, glycogen synthesis ↑	STZ-induced Kunming mice (*n* = 10)	150–1200 mg/kg/day	21 days	Catechin, epicatechin, procyanidins, gallic acid, protocatechuic acid, ferulic acid, rutin, isoquercitrin, quercitrin, anthocyanin, anthocyanidin, leucoanthocyanidins, triterpenoids	[208,209,210,211,212,616]
15. *Azadirachta indica*	Neem	Leaves, stem, bark, flower, roots, fruit	Fever, skin diseases, infection, inflammation and dental disorders	PPBG↓, FBG↓, HbA1c↓, IR↓, endothelial function↑, oxidative stress ↓, systemic inflammation ↓	T2DM patients (30–65 years old, *n* = 20)	125–500 mg/kg/twice a day	84 days	Nimbidin, nimbin, nimbidol, quercetin nimbosteron, saponin, tannin, flavonoids	[213,214,215,216,217]
16. *Beta vulgaris*	Beetroot	Fruit	Dandruff, loss of libido, stomachaches, diabetes, arthritis, constipation	Blood glucose↓, HbA1c↓, FBG↓, TC↓, TG↓, LDL-C↓, IR↓, HDL↑, ALT↓, AST↓, gluconeogenesis↓, α-amylase and α-glucosidase activity↓	T2DM patients (57 ± 4.5 years, *n* = 44)	100,000 mg/kg/day	56 days	Betalains, betanin, carotenoids, coumarins, sesquiterpenoids, betagarin, betavulgarin, quercetin, kaempherol, tiliroside, astragalin, rhamnocitrin, rhamnetin, betavulgarosides, betacyanin	[218,219,220,617]
17. *Brassica juncea*	Mustard	Seeds	Arthritis, foot-ache, lumbago, diabetes, rheumatism	Blood glucose↓, FBG↓, TC↓, TG↓, prediabetic IR↓, glucose tolerance↑, insulin secretion↑, intestinal glucose absorption↓	Fructose-induced Sprague Dawley rats (*n* = 6)	100 mg/kg/day	30 days	Chlorogenic acid, sinigrin, *p*-coumaric acid, vanillic acid, flavonoids, chlorogenic acid, polyphenols, allyl isothiocyanate, cinnamic acid, kaempferol	[221,222,223,224]
18. *Brassica oleracea* var. capitata	Cabbage	Flower	gastritis, peptic ulcers, irritable bowel syndrome, diabetes, idiopathic cephalalgia	FBG↓, TC↓, TG↓, LDL-C↓, HDL↑, insulin sensitivity↑, β-cell function↑	Alloxan-induced diabetic rabbits (*n* = 7)	500 mg/kg/day	30 days	Myricetin, quercetin, kaempferol, apigenin, luteolin, cyanidin daidzein, genistein, glycitein, biochanin A, formononetin	[225,226,227]
19. *Brassica oleracea* var. italica	Broccoli	Flower	Xerophthalmia, hyperlipidemia, fibromyalgia, cancer, diabetes	Blood glucose↓, lipid peroxidation↓, IL-6↓, TNF-α↓, HbA1c↓, insulin sensitivity↑, β-cell function↑, glucose production ↓.	T2DM Albino Wistar Rats (*n* = 8)	400 mg/kg/day	42 days	Glucosinolates, isothiocyanates, sulforaphane, sinapic acid, gallic acid, vanillic acid, *p*-coumaric acid, ferulic acid, chlorogenic acid, apigenin, kaempferol, luteolin, quercetin, and myricetin	[228,229,618]
20. *Camellia sinensis*	Tea	Leaves	Flatulence, indigestion, vomiting, obesity, diarrhea, hyperglycemia, stomach discomfort	Blood glucose↓, IR↓, MDA↓, oxidative stress, inflammatory cytokines↓, α-amylase and α-glucosidase activity↓, insulin release ↑, glycation ↓, glucose tolerance↑	STZ-induced Wistar rats (*n* = 8)	100–200 mg/kg/day	28 days	Caffeine, theanine, proanthocyanidins, myricetin, kaempferol, quercetin, chlorogenic acid, coumarylquinic acid, theogallin, catechins, epicatechin	[230,231,232,233,619]
21. *Capsicum annuum* L.	Red pepper	Seeds	Dyspepsia, ulcer, anorexia, GERD and diabetes.	FBG↓, HbA1c↓, inflammatory cytokines↓, TG↓, TNF-α↓, IL-6↓, plasma insulin↑, gluconeogenesis↓, AMPK↑, FOXO1↑, glucose uptake↑, GLUT-4 translocation↑	High fat died induced C57BL/KsJ (*n* = 8)	200 mg/kg/day	56 days	Lycopene, flavonoids, carotenoids, flavones, apigenin, quercetin, isoquercetin, capsinoids, polyphenols	[234,235,236,237,238]
22. *Carica papaya*	Papaya	Fruit, seeds, leaves	Hypertension, fever (dengue), obesity, jaundice, UTI, ulcer, constipation, bronchitis, cough, diarrhea, asthma, piles, malaria, wound healing	Blood glucose↓, TG↓, TC↓, α-amylase and α-glucosidase activities↓, oxidative stress ↓	STZ-induced Wistar rats (*n* = 6)	750–3000 mg/100 mL/day	28 days	Papain, quercetin, kaempferol, *p*-coumaric acid, carpinine, carpaine, choline, β-carotene, linalool, oleic acid, linolenic acid	[239,240,241,242]
23. *Carissa carandas*	Bengal currant	Fruits	Anorexia, brain disease, cough, asthma, constipation, diarrhea, diabetes, pain, pharyngitis, scabies, leprosy, malaria, myopathic spams, fever, epilepsy, seizures	Blood glucose↓, inflammation↓, α-amylase and α-glucosidase activity↓	Alloxan-induced albino rats (*n* = 5)	400 mg/kg	1 day	Lignans, flavonoids, steroids, phenolic acids, alkaloids	[243,244,245,246,247]
24. *Catharanthus roseus* L.	Vinca Rosea	Flowers, leaves	Cancer, diabetes, stomach disorders, kidney, liver, cardiovascular disorders	Blood glucose↓, insulin secretion↑, β-cell function↑, TC↓, creatinine↓	Alloxan- induced Albino rabbits (*n* = 5)	0.5–1 mg/kg/day	24 h	Gallic acid, rutin, *p*-coumaric acid, ajmalicine, vindoline, catharanthine, vinblastine, vincristine, caffeic acid, quercetin, kaempferol, syringic acid, chlorogenic acid, ellagic acid, coumarins	[248,249,250,251,252]
25. *Centella asiatica*	Centella leaves	Leaves	Leprosy, lupus, varicose ulcers, eczema, psoriasis, diarrhea, fever, amenorrhea, female genitourinary tract infections, diabetes, anxiety	Blood glucose↓, insulin sensitivity↑, oxidative stress↓, inflammation↓	STZ-induced Sprague-Dawley rats (*n* = 6)	500–1000 mg/kg/day	14 days	Asiaticoside, madecassic acid, madecassoside, centellase, quercetin, kaempferol, phytosterol	[253,254,255,256]
26. *Chenopodium quinoa*	Quinoa	Grains	Dyslipidemia, diabetes, heart disease	Blood glucose↓, FBG↓, IR↓, TC↓, TG↓, LDL-C↓, α-glucosidase activity↓, lipid accumulation↓, glucose tolerance↑, insulin sensitivity↑	High fat diet induced C57BL/6J mice (*n* = 6)	2000 mg/kg/day	84 days	Saponins, phytosterols, phytoecdysteroids, phenolics, tocophenols, betalains, tannins, glycine betaine	[257,258,259,260,261,262,263]
27. *Cicer arietinum*	Chickpea	Grains	Digestive diseases, cancer, cardiovascular disease, diabetes	Blood glucose↓, inflammation↓, organ function↑, intestinal dysbiosis↓, α-amylase, α-glucosidase and DPP4 activity↓, carbohydrate metabolism↑, body weight↓	STZ-induced HFF rats (*n* = 7)	3000 mg/kg/day	28 days	Uridine, adenosine, tryptophan, 3-hydroxy-olean-ene, biochanin	[264,265,266,267,268,269,620]
28. *Cinnamomum verum*	Cinnamon	Bark	Nausea, vomiting, fever, halitosis, arthritis, coughing, hoarseness, frigidity, cramps, intestinal spasms, bronchitis, asthma, odontalgia, cardiac diseases, diarrhea, vaginitis, neuralgia, rheumatism, piles, urinary disease	Blood glucose↓, GLUT-4 translocation↑, glucose uptake↑, Mitochondrial UCP-1↑, insulin secretion↑, α-glucosidase activity↓,	STZ-induced Wistar rats (*n* = 20)	30 mg/kg/day	22 days	cinnamaldehyde, cinnamates, cinnamic acid, eugenol, cinnamyl acetate, cubebene, terpinolene, linalool, linalyl acetate, benzyl cinnamate, piperitone, β-sitosterol, flavanol, glucosides, coumarin, protocatechuic acid, vanillic acid, syringic acid	[270,271,272,273]
29. *Citrullus lanatus* L.	Water-melon	Fruit, seeds	Gastrointestinal disorders, urinary disorders, aphrodisiac, fever, laxative, emetic	FBG↓, serum lipid profile↓, glucose-6-phosphatase↓, lipid peroxidation↓, GLUT4↑, GLUT2↑, hexokinase activity↑	Alloxan-induced Wistar Albino rats (*n* = 3)	500–1000 mg/kg/day	14 days	Stigmasterol, quinic acid, malic acid, epicatechin, caffeic acid, rutin, *p*-coumaric acid, quercetin, ferulic acid, scopoletin, apigenin, kaempferol, β carotene, citrulline, lycopene, α tocopherol	[274,275,276,277]
30. *Citrus limon*	Lemon	Fruit, peel, leaves	Cough, scurvy, cold, fever, rheumatism, sore throat, diabetes, irregular menstruation	Serum glucose↓, body weight↓, TC↓, TG↓, LDL↓, VLDL↓, GSH↑, insulin sensitivity↑, GLUT-4 translocation↑, AGE formation↓, Glucose uptake↑	STZ-induced Wistar rats (*n* = 6)	200–400 mg/kg/day	15 days	Limocitrin, hesperidin, diosmin, hesperetin, didymin, naringin, naringenin, tangeretin, rutine, quercetin, β-pinene, γ-terpinene, D-limonene, ferulic acid	[278,279,280,281,282,283,284,285]
31. *Citrus maxima*	Pomelo	Fruit, peel	Asthma, fever, ulcer, diarrhea, cough, Alzheimer’s disease, diabetes, insomnia	Blood glucose↓, TG↓, TC↓, HDL↑, LDL↓, α-amylase, α-glucosidase and angiotensin I-converting enzyme activity↓, body weight↓, glucose tolerance ↑	Alloxan- induced diabetic rats (*n* = 7)	200–600 mg/kg/day	14 days	Terpenoids, sterols, carotenoids, polyphenols, chlorogenic acid, ferulic acid, caffeic acid, gallic acid, ρ-coumaric acid.	[286,287,288,621]
32. *Citrus reticulata*	Orange	Fruit, peel	Alzheimer’s disease, cough, phlegm, diabetes, hepatic steatosis, cancer	mRNA expression ↑, GLUT-4 translocation↑, insulin sensitivity↑, serum fructosamine level ↓, glucose tolerance↑	STZ-induced Wistar rats (*n* = 6)	100 mg/kg/day	28 days	Flavonoids hesperidin, quercetin, naringin, nobiletin, tangeretin	[289,290,291,292,293]
33. *Cocos nucifera*	Coconut	Fruit, husk, water	Diarrhea, diabetes, dermatitis, renal diseases, stomachaches, fever, asthma, abscesses, amenorrhea, gonorrhea, menstrual disorders	Blood glucose↓, α-amylase and α-glucosidase activity ↓, DPPH free radicals↓, IR↓, oxidative stress↓, neuropathy↓, β-cell regeneration ↑	STZ-induced Wistar rats (*n* = 6)	250–500 mg/kg/day	28 days	Chlorogenic, gallic, ferulic, salicylic, coumaric acids, glycosides, rutin, quercetin, vanillin, catechin, epicatechin, neochlorogenic acid, chlorogenic acid, lutein	[295,296,297,298,299,300,301,302]
34. *Coffea Arabica* L.	Coffee	Leaves, fruit, beans	Flu, anemia, edema, asthenia., asthma, backache, cough, jaundice, diarrhea, intestinal pain, migraine, headache, fever, purulent wounds, pharyngitis, diabetes, stomatitis	Blood glucose↓, insulin secretion↑, α-amylase and α-glucosidase activity↓, nephropathy↓, plasma insulin↑, IR↓, TG↓	STZ-induced Wistar rats (*n* = 6–8)	1000 mg/kg/day	90 days	Chlorogenic acids, caffeic, *p*-coumaric, vanillic, ferulic, protocatechuic acids, flavonoids, alkaloids, caffeine, sitosterol, stigmasterol, coffeasterin, kaempherol, quercetin, sinapic, quinolic, trigonelline, caffeoylquinic, dicaffeoylquinic	[303,304,305,622]
35. *Colocasia esculenta*	Taro	Stem, leaves	Rheumatic pain, diabetes, hypertension, pulmonary congestion	Blood glucose↓, HbA1c↓, TC↓, TG↓, LDL-C↓, VLDL↓, HDL↑, body weight↓	STZ-induced Wistar rats (*n* = 6)	405–810 mg/kg/day	28 days	Tannins, phytates, oxalates, tryptophan, chlorogenic acid, anthraquinone, vitexin, catechins, apigenin, cinnamic acids, isovitexin, orientin, isoorientin, rosmarinic acid	[306,307,308,309,310]
36. *Coriandrum sativum*	Coriander	Seeds, leaves	Diarrhea, flatulence, colic, indigestion, gastrointestinal diseases, diabetes	Diabetic neuropathy↓, Blood glucose↓, MDA↓, GSH↑, SOD↑, TC↓, TG↓, LDL-C↓, AGEs formation↓, lipid peroxidation↓, oxidative stress↓, TNF-α↓	STZ-NAD induced Wistar rats (*n* = 6)	100–400 mg/kg/day	45 days	Flavonoid, tocopherol, tocotrienol sterol, carotenoids, terpenoids, steroids, saponin, tannin, alkaloids	[302,311,312,313,314]
37. *Crocus sativus* L.	Saffron	Flower stigma	CNS diseases, diabetes, obesity, cancer, dyslipidemia	Blood glucose↓, MDA↓, NO↓, GSH↑, SOD↑, TC↓, TG↓, LDL-C↓, α-amylase and α-glucosidase activity↓, inflammation↓	STZ-induced Wistar rats (*n* = 9)	10–40 mg/kg/day	28 days	Crocin, β carotenes, crocetin, picrocrocin, zeaxanthene, safranal	[315,316,317,318,319,320]
38. *Cuminum Cyminum* L.	Cumin seeds	Seeds	Diarrhea, dyspepsia, epilepsy, toothache, whooping cough, flatulence, indigestion, diabetes, jaundice	Blood glucose↓, AGEs formation↓, HbA1c↓, creatinine↓, blood urea nitrogen↓, serum insulin↑, oxidative stress↓, nephropathy↓	STZ-induced Wistar rats (*n* = 6)	200–600 mg/kg/day	28 days	Carvacrol, carvone, α-pinene, limonene, γ-terpinene, linalool, carvenone, *p*-cymene, cumin aldehyde, limonene, α- and β-pinene, terpinenes, safranal, and linalool	[321,322,323]
39. *Cucumis sativus*	Cucumber	Fruit, seeds	Sunburn, skin irritation, constipation, thermoplegia, gall bladder stone, hyperdipsia, diabetes	Blood glucose↓ IR↓, body weight↓, insulin sensitivity↑, gluconeogenesis↓, glucagon secretion↓	STZ-induced Wistar rats (*n* = 6)	200–800 mg/kg/day	9 days	Cucurbitacin, cucumerin, cucumegastigmanes vitexin, orientin, apigenin, isoscoparin	[324,325,326,327]
40. *Cucurbita pepo* L.	Pumpkin	Fruit, seeds	Dermatitis, depression, irritable bladder, intestinal inflammation, prostate enlargement, hyperglycemia	Blood glucose↓, TC↓, TG↓, LDL-C↓, HDL↑, IR↓, ROS↓, SOD↑, GSH↑, MDA↓	STZ-induced T2DM mice (*n* = 10)	400 mg/kg/day	56 days	β-carotene, zeaxanthin, lutein, flavonoids, alkaloids, polysaccharides, polyphenols	[328,329,330,331,332]
41. *Curcuma longa* L.	Turmeric	Fruit	Cough, diabetes, arthritis, gall bladder stones, dermatitis, cancer, intestinal, stomachic diseases	Blood glucose↓, FBG↓, insulin sensitivity↑, β-cell function↑, IR↓, GLUT-2 activity↑, insulin secretion↑, glucose uptake↑	STZ-NA induced Wistar rats (*n* = 6)	30–60 mg/kg/day (*n* = 6)	30 days	Caffeic acid, curdione, coumaric, caffeic acid, casuarinin, curcuminol, isorhamnetin, valoneic acid, eugenol, corymbolone, demethoxycurcumin	[333,334,335,336,337]
42. *Daucus carota*	Carrot	Fruit	Diarrhea, constipation, intestinal inflammation, weakness, illness, diabetes, rickets	Blood glucose↓ IR↓, Obesity↓, body weight↓, BMI↓, α-amylase and α-glucosidase activity↓	High fructose induced Wistar rats (*n* = 6–14)	50 mL/kg/ day	56 days	Carotenoid, polyacetylenes, ascorbic acid, α and β-carotene, lutein, lycopene, anthocyanins	[338,339,340,341,623]
43. *Ficus carica*	Fig	Fruit, leaves, bark, roots	Dermatitis, leprosy, cancer, anemia, diabetes, paralysis, urinary tract infection, ulcer, liver diseases	FBG↓, PPBG↓, TG↓, HDL↑, LDL↓, VLDL↓, TC↓, pancreatic β-cell apoptosis↓, pancreatic AMPK↑, caspase-3↓, body weight ↓	STZ-induced C57BL/6 mice (*n* =12)	2000 mg/kg/day	42 days	Eugenol, anthocyanins, volatile compounds, phenolic acids, flavones, flavanols	[342,343,344,345,346,347]
44. *Fragaria ananassa*	Strawberry	Fruit, leaves	Wound healing, platelet aggregation, obesity, diabetes	Blood glucose↓, IR↓, insulin secretion↑, α-amylase and α-glucosidase activities↓, plasma creatinine↓, MDA↓, TNF-α↓, IL-6↓, caspase-3↓	STZ-induced Albino rats (*n* =4)	50–200 mg/kg/day	30 days	Quercetin, kaempferol, rutin, gallic acid, chlorogenic acid, caffeic acid, ellagitannins, octadecatrienoic acid, vitamin C and E, folic acid, carotenoids, anthocyanins, gallotannins	[263,348,349,350,351,624]
45. *Glycine max*	Soya bean	Seeds, leaves	Osteoporosis, cardiovascular disease, diabetes	Blood glucose↓, FBG↓, IR↓, TC↓, TG↓, LDL-C↓, α-glucosidase activity↓, HbA1c↓, HDL↑, body weight↓, glucose uptake↑	T2DM obese patients (43–51 years, *n* = 15)	2000 mg/kg/day	84 days	β-conglycinin, phenolic acids, flavonoids, isoflavones, saponins, phytosterols, sphingolipids	[352,353,354,355,356,625]
46. *Helianthus* annuus	Sunflower	Flowers, seeds	Diabetes, nephrotoxicity, cardiovascular disease, hematologic disorders	Blood glucose↓, nephropathy↓, FBG↓, BMI↓, body weight↓, AGEs formation↓, DPPH↓, NO↓, urea↓	Alloxan- Induced Albino rats (*n* = 6)	150–600 mg/kg/day	21 days	Flavonoids, alkaloids, saponins, tocopherols, carotenoids, saponins, tannins, chlorogenic acid, and caffeic acid	[357,358,359,360]
47. *Hibiscus rosa-sinensis* Linn.	China rose	Flowers, leaves	Tumor, hairloss, infertility, diabetes, wounds	Blood glucose↓, insulin secretion↑, β-cell function↑, TC↓, TG↓, hepatic glycogen↓, SOD↑	STZ-induced Long Evans rats (*n* = 6–8)	250–500 mg/kg/day	28 days	Quercetin, cyanidin, ascorbic acid, genistic acid, lauric acid, thiamine, niacin, margaric acid, calcium oxalate, hentriacontane	[361,362,363,364,365]
48. *Hylocereus undatus*	Dragon fruit	Fruit, seeds	Diuretic, healing agent, laxative, gastritis aid	Blood glucose↓, MDA↓, FBG↓, SOD↑, GLUT2↑, oxidative stress↓	STZ-induced Sprague Dawley rats (*n* = 6)	250–500 mg/kg/day	35 days	Lycopene, β-carotene, betacyanin, oleic acid, octacosane, phthalic acid, eicosane, tetratriacontane, tacosane, campesterol linoleic acid, palmitic acid, gallic acid, syringic acid, protocatechuic acid, *p*-coumaric acid	[366,367,368,626]
49. *Ipomoea batatas*	Sweet potato	Fruit	Aphrodisiac, burns, catarrh, diarrhea, fever, nausea, splenosis, stomach distress, anemia, tumors, hypertension, prostatitis, asthma,	Blood glucose↓, IR↓, Insulin sensitivity↑, glucose tolerance↑, insulin secretion↑	T2DM patients (58 ± 8 years, *n* = 6)	4000 mg/kg/day	42 days	Anthraquinones, coumarins, flavonoids, saponins, tannins, phenolic acids, quercetin, chlorogenic acid, terpenoids, β-carotene, zeaxanthin, lutein, anthocyanins	[369,370,371,372,373,627]
50. *Juglans regia* L.	Walnut	Nut, leaves	Curing bacterial infections, stomachaches, thyroid issues, diabetes. cancer, heart conditions, sinusitis	Blood glucose↓, α-amylase and α-glucosidase activity↓, PTP1B↓	STZ-induced Wistar rats (*n* = 7)	25–100 mg/kg/day	28 days	tocopherol, gallic acid, protocatechuic acid, caffeic acid, chlorogenic acid, catechin, vanillic acid, epicatechin, *p*-coumaric acid, isoquercitrin, quercetin, luteolin, kaempferol, and apigenin	[374,375,376,377]
51. *Lactuca sativa*	Lettuce	Leaves	Hyperglycemia, osteodynia, inflammations	FBG↓, TC↓, TG↓, LDL-C↓, HDL↑, β-cell function↑, SOD↑, GSH↑, glucose production ↑	STZ-induced Wistar rats (*n* =10)	50 mg/kg/day	28 days	flavonoids, quercetin, flavonols, anthocyanins, hydroxycinnamoyl derivatives	[378,379,380,381,382]
52. *Lagenaria siceraria*	Bottle gourd	Fruit, leaves, seeds	Jaundice, diabetes, constipation, flatulence, insomnia, ulcer, piles, colitis, insanity, hypertension, congestive cardiac failure, skin diseases, headaches	Blood glucose↓, HbA1c↓, FBG↓, body weight ↓, TC↓, TG↓, insulin production↑, glucose tolerance↑, intestinal glucose absorption↓	STZ-induced Wistar rats (*n* = 6)	200–400 mg/kg/day	15 days	Isovitexin, isoorientin, saponarin, fucosterol, campesterol, cucurbitacin B, cucurbitacin D, cucurbitacin E, isoquercitrin, kaempferol, gallic acid, and protocatechuic acid	[383,384,385,386]
53. *Laurus nobilis*	Bay leaves	Leaves	Stomachaches, phlegm, cold, sore throat, headache, indigestion, flatulence, eructation, epigastric bloating, diabetes	Blood glucose↓, β-cell function↑, α-glucosidase activity↓, Insulin production↑, β-cell regeneration↑	STZ-induced Wistar rats (*n* = 6)	200 mg/kg/day	28 days	Kaempferol, syringic acid, quercetin, apigenin, luteolin, lauric acid, palmitic acid, linoleic acid, lutein, eugenol	[387,388,389,390]
54. *Litchi chinensis*	Lychee	Fruit, seeds	Cough, ulcer, flatulence, testicular swelling, diabetes, hernia, obesity	Blood glucose↓, FBG↓, renoprotection↑, IR↓, glucose tolerance↑, TG↓, α-glucosidase activity↓	Alloxan- induced Wistar rats (*n* = 11)	2.6 mg/kg/day	30 days	Flavonoids, triterpenes, sterols, phenolic compounds	[391,392,393]
55. *Luffa acutangula*	Ridge gourd	Fruit, seeds	Jaundice, hemorrhoids, dysentery, headache, ringworm infection, insect bite, urinary bladder stone, granular conjunctivitis, constipation, leprosy, diabetes	Blood glucose↓, HbA1c↓, FBG↓, ALT↓, AST↓, TC↓, TG↓, LDL-C↓, VLDL↓, gluconeogenesis↓	STZ-induced Wistar rats (*n* = 6)	100–400 mg/kg/day	21 days	Luffaculin, luffangulin, apigenin, luteolin, myristic acid, palmitic acid, oleic acid, linoleic acid, oleanolic acid, machaelinic acid, α-thujene, terpinene	[394,395,628]
56. *Malus domestica Borkh*	Apple	Fruit, peel	Wound healing, diabetes, asthma, obesity, cardiovascular disease	Blood pressure↓, endothelial function↑, lipid homeostasis↑, insulin resistance ↓	HFHF-fed ICR mice (*n* = 8)	250 mg/kg/day	28 days	Procyanidins, flavonoids, chlorogenic acids, hydroxycinnamic acids, anthocyanins, quercetins	[396,397,398,399,400,401,402,403,404,405,629]
57. *Mangifera indica*	Mango	Fruit, peel, bark, seeds	Asthma, tetanus, polyuria, dysentery, anthrax, indigestion, tumor, tympanites, diarrhea, colic	FBG↓, HbA1c↓, serum fructosamine level↓, plasma insulin ↑, α-amylase and α-glucosidase activities↓, PPBG ↓	STZ-induced Wistar rats (*n* = 6)	100–200 mg/kg/day	60 days	Mangiferins, carotenoids, flavonoids, anthocyanins, gallic acid, protocatechuic acid, chlorogenic acid, ferulic acid	[406,407,408,409,410,411]
58. *Mentha spicata*	Mint leaves	Leaves	Cough, cold, asthma, fever, obesity, dementia, hypertension, abdominal pain, headache, menstrual pain, depression, insomnia	FBG↓, TC↓, TG↓, LDL-C↓, VLDL↓ MDA↓, body weight↓, HDL↑, α-amylase and α-glucosidase activity↓	Alloxan-induced Wistar rats (*n* = 6)	300 mg/kg/day	21 days	Carvone, limonene, 1,8-cineole, pulegone, β-bourbonene, β-pinene, dihydrocarveol, α-phellandrene, borneol, linalool, germacrene D, and piperitone	[412,413,414,630]
59. *Moringa oleifera* Lam.	Moringa	Fruit, leaves	Diabetes, liver disease, cancer, inflammation, hypercholesteremi, hypertension	Blood glucose↓, hepatic functions↑, FBG↓, TC↓, TG↓, LDL-C↓, VLDL↓, HDL↑, α-amylase and α-glucosidase activity↓	High fat died induced C57BL/6 mice (*n* = 6)	200 mg/kg/day	21 days	Tannins, βcarotene, vitamin C, quercetin, alkaloids, saponins, steroids, phenolic acids, glucosinolates, flavonoids, terpenes	[415,416,417,418,419]
60. *Momordica charantia*	Bitter gourd	Fruit, leaves, seeds	T2DM, dyslipidemia, cancer, obesity, malaria, dysentery, hypertension, worm infections	Blood glucose↓, fructosamine↓, IR↓, TC↓, TG↓, insulin secretion↑, HDL↑, MDA↓, GSH↑, glucose uptake↑, β-cell function↑	STZ-induced Wister rats (*n* = 6–8)	10 mL/kg/day	21 days	Saponins, triterpenes, flavonoids, ascorbic acids, steroids, tannins, alkaloids, cardiac glycosides, phlobatinnins, anthraquinones	[420,421,422,423,424,425,426,427]
61. *Morus alba* L.	Mulberry	Fruit, leaves	Insomnia, tinnitus, dizziness, premature aging, diabetes	FBG↓, IR↓, TG↓, HDL↑, LDL↓, TC↓, GLUT-4 translocation↑	STZ-induced HFF Wistar rats (*n* = 6)	400 mg/kg/day	49 days	Quercetin, isoquercetin alkaloids, polyphenols, flavonoids, anthocyanins	[429,430,431,432]
62. *Murraya koenigii* L.	Curry leaves	Leaves	Piles, inflammation, itching, fresh cuts, dysentery, bruises, edema, body aches, diabetes, snakebites	Blood glucose↓, MDA↓, GSH↑, IR↓, β-cell regeneration↑	STZ-NA induced Sprague Dawley rats (*n* = 5)	200–400 mg/kg/day	28 days	Mahanine, mahanimbine, murrayanol, koenimbine, koenigicine, koenigine, murrayone, isomahanine, glycozoline, mukonicine, murrayazolinol, murrayacine, quercetin, apigenin, kaempferol, catechin	[433,434,435,436]
63. *Myristica fragrans* Houtt.	Nutmeg	Fruit, seeds	Skin infection, diarrhea, diabetes, Alzheimer’s diseases, rheumatism, asthma, cold, cough, malaria	Blood glucose↓, serum insulin↑, oxidative stress↓, β-cell function↑, AMPK↑, IL-6↓, TNF-α↓	Chlorpromazine-induced obese Swiss albino mice (*n* = 4–6)	50–450 mg/kg/day	7 days	Flavonoids, terpenes, phenylpropanoids, coumarin, lignans, alkanes, and indole alkaloids	[437,438,439,440]
64. *Nigella sativa* L.	Black seeds	Seeds	Asthma, dyslipidemia, diabetes, diarrhea	Blood glucose↓, α-amylase and α-glucosidase activity↓, serum lipids↓ insulin sensitivity↑, gluconeogenesis↓	STZ-induced Swiss albino mice (*n* = 6)	100–700 mg/kg/day	28 days	Thymoquinone, thymol, limonene, carvacrol, *p*-cymene, longifolene, α-pinene, linoleic acid, oleic acid, palmitic acid, saponins, flavonoids, alkaloids	[441,442,443,444,445]
65. *Ocimum sanctum* L.	Holy basil	Leaves, seeds	Anxiety, cough, asthma, diarrhea, fever, dysentery, arthritis, eye diseases, skin diseases, malaria, vomiting, cardiac and genitourinary infection	TC↓, TG↓, LDL↓, VLDL↓, atherogenic index ↓, GSH ↑, Insulin production↑, intestinal glucose absorption↓	Alloxan-induced diabetic rabbits (*n* = 5)	0.8 mg/kg/day	28 days	Eugenol, euginal, urosolic acid, carvacrol, linalool, caryophyllene, triterpenoids, tannins	[446,447,448,449,631]
66. *Olea europaea* L.	Olive	Fruit, leaves	Diabetes, diarrhea, inflammation, urinary tract infection, intestinal diseases, hemorrhoids, rheumatisms	Blood glucose↓, inflammatory cytokines↓, body weight↓, gluconeogenesis↓, glucose-6-phosphatase enzyme activity↓	HFF-STZ-induced Wistar rats (*n* = 5)	200–400 mg/kg/day	70 days	Flavonoids, secoiridoids, hydroxytyrosol and tyrosol, cinnamic acid	[450,451,452,453,454,455]
67. *Origanum vulgare*	Oregano	Leaves	Acne, cystic fibrosis, diabetes, bacterial infections	Blood glucose↓, glucose uptake↑, GLUT2↑, α-amylase and α-glucosidase activity↓, oxidative stress↓	STZ-induced Diabetic rats (*n* = 6)	20 mg/kg/day	15 days	Amburoside, apigenin, luteolin 7-*O*-glucuronide, rosmarinic acid, and lithospheric acid	[456,457,458,459,632]
68. *Passiflora edulis*	Passion fruit	Fruit, peel	Cough, diabetes, dysmenorrhea, dysentery, arthralgia, constipation	Blood glucose↓, TG↓, TC↓, interleukins↓, body weight↓, insulin sensitivity ↑, glucose tolerance ↑	Cafeteria diet induced C57BL/6 mice (*n* = 10)	15% of PEPF (P. edulis peel flour) in CAF diet	112 days	Piceatannol, flavonoids, triterpenoids, tocopherols, linoleic acid, vitexin, carotenoid, orientin, isoorientin, gallic acid, rutin, quercetin, ascorbic acid	[460,461,462,463,464,465,466,467,468]
69. *Persea americana Mill.*	Avocado	Fruit, leaves, seeds, bark	Cardiovascular diseases, diabetes	Blood glucose↓, metabolic state ↑, activation of Akt/Pkb, glucose uptake↑, β-cell regeneration↑, HDL-c↑, LDL↓	STZ-induced Wistar rats (*n* = 7)	150–300 mg/kg/day	28 days	Flavonoids, alkaloids, saponins, tannins, carbohydrates, glycosides	[469,470,471,472,473]
70. *Petroselinum crispum*	Parsley	Leaves, seeds, roots	Otitis, urinary tract infection, dysmenorrhea, hypertension, diabetes, dermatitis, gastrointestinal disorders	Blood glucose↓, NEG↓, lipid peroxidation↓, body weight↓, GSH↓, insulin sensitivity↑, gluconeogenesis↓	STZ-induced Swiss albino rats (*n* = 13–20)	2000 mg/kg/day	42 days	Courmarins, phthalides, phenyl propanoids, tocopherols, apigenin, myristicin, apiol	[474,475,476,477]
71. *Phaseolus vulgaris* L.	Kidney bean	Seeds	Wound healing, pharyngitis, fever, unpleasant body odor, obesity, diabetes, vaginal infection	Blood glucose↓, insulin sensitivity↑, TC↓, TG↓, gluconeogenesis↓, α-glucosidase activity↓	STZ-induced Wistar rats (*n* = 5)	150 mg/kg/day	40 days	Protocatechuic acid, *p*-coumaric acid, procyanidin, myricetin, naringenin, gallic acid, quercetin, catechin, kaempferol, ferulic acid	[478,479,480,481]
72. *Phoenix dactylifera* L.	Date	Fruit, leaves	Fever, inflammation, nervous disorders, loss of consciousness, dementia	Blood glucose↓, serum insulin↑, MDA↓, TNF-α↓, CRP↓	STZ-induced diabetic rats (*n* =10)	200 mg/kg/day	30 days	Ellagic acid, gallic acid, *p*-coumaric acid, apigenin, naringin, gallic acid, catechin, ferulic acid, sinapic acid, epicatechin, vanillic acid, coumarin, quercetin, rutin, myricetin, luteolin, kaempferol, isorhamnetin, rhamnetin, β-sitosterol, isorhamnetin, procyanidin, protocatechuic acid	[482,483,484,485,633]
73. *Phyllanthus emblica* L.	Amla	Fruit, leaves, bark, roots	Cold, fever, cough, hyperacidity, peptic ulcer, erysipelas, jaundice, diarrhea, dysentery, leprosy, hemorrhages, hematogenesis, anemia, asthma, bronchitis, colic, dyspepsia, hepatopathy, leucorrhea, menorrhagia	Blood glucose↓, TG↓, TC↓, LDL↓, HDL↑, α-amylase and α-glucosidase activities↓, AMPK↑	STZ-induced Wistar rats (*n* = 6)	25–75 mg/kg/day	28 days	Phyllembelic acid, gallic acid, ellagic acid, pectin, quercetin, linolenic, linoleic, oleic, stearic, palmitic, myristic acid, tannins, chebulic, chebulagic, chebulinic acids, alkaloids phyllantidine, phyllantine, lupeol, leucodelphinidin. corilagin, digallic acid, kaempferol, and zeatin	[486,487,488,489,634]
74. *Piper betle* L.	Betel leaf	Leaves	Wound healing, bronchitis, diabetes, cough, indigestion in children, headaches, arthritis,	FBG↓, HbA1c↓, IR↓, insulin production↑, glucokinase activity↑	STZ-induced Wistar rats (*n* = 6)	75–150 mg/kg/day	30 days	Estragole, linalool, safrol, terpenes, phenols, steroids, saponins, tannins	[490,491,492,493]
75. *Pisum sativum* L.	Pea	Seeds	Blood purifying, wrinkled skin, acne, phlegm, intestinal inflammation, constipation, diabetes	Blood glucose↓, HbA1c↓, NO↓, plasma insulin ↑, glucose homeostasis↑, glucose tolerance↑, polyphagia↓, TG↓, LDL-C↓, α-glucosidase activity↓, body weight↓	STZ-induced ICR mice (*n* = 6)	100–400 mg/kg/day	42 days	Flavonoid, quercetin, ellagic acid, coumaric acid, β-sitosterol, β-amyrin, catechin, myricetin, vanillic acid, kaempferol	[494,495,496,497]
76. *Prunus armeniaca* L.	Apricot	Fruit, leaves	Cancer, atherosclerosis, angina, retinopathy, nephropathy, hypertension, diabetes	Blood glucose↓, FBG↓, α-glucosidase activity↓, HbA1c↓, insulin secretion↑, oxidative stress ↓	Alloxan-induced Swiss mice (*n* = 7)	2–8 mg/kg/day	56 days	Chlorogenic, gallic, ferulic, salicylic, coumaric acids, glycosides, rutin, quercetin, vanillin, catechin, epicatechin, neochlorogenic acid, chlorogenic acid, lutein	[498,499,500]
77. *Prunus domestica*	Plum	Fruit	Anemia, neurasthenia, leukorrhea, Alzheimer’s disease, irregular menstruation, anxiety, diabetes, constipation	Blood glucose↓, TG↓, TC↓, LDL↓, α-amylase and α-glucosidase activities↓, HMGCoA reductase↓, oxidative stress ↓	STZ-induced Swiss Albino mice (*n* =10)	50 mg/kg/day	20 days	Chlorogenic acid, neochlorogenic acid, tocopherols, β-carotenes, quercetin, myricetin, kaempferol, citric acid, malic acid	[501,502,503,504,505,506,507,508]
78. *Prunus dulcis*	Almonds	Nut	CNS disorders, respiratory disorders, diabetes, urinary tract infections	FBG↓, TC↓, TG↓, LDL↓, stomach emptying, time↓, insulin production↑	T2DM patients (58 ± 2 years, *n* = 20)	60,000 mg/kg/day	84 days	Oleic acid, linoleic acid, palmitic acid, arachidic acid, anthocyanin, kaempferol, quercetin, isorhamnetin, galactosidase, chlorogenic acid	[509,510]
79. *Prunus persica* L.	Peach	Fruit, peel, leaves	Enhancing blood circulation, blood clotting, constipation, diabetes	Body weight↓, lipid metabolism↑, lipogenesis↓, fatty acid oxidation↑, α-amylase and α-glucosidase activities↓, β-cell regeneration↑	HFF C57BL/6 male mice (*n* = 12)	200–600 mg/kg/day	56 days	Procyanidin, epicatechin, catechin, prunin, phloridzin, naringenin, neochlorogenic acid, caffeoylquinic acid, chlorogenic acid, quercetin, aucubin, kaempferol, prunitrin	[511,512,513,514,635]
80. *Punica granatum*	Pome-granate	Fruit, peel, seeds	Dysentery, diarrhea, piles, bronchitis, biliousness, diabetes	Blood glucose↓, TG↓, TC↓, HDL↑, LDL↓, intestinal glucose absorption↓, GLUT-4 translocation ↑	Alloxan-induced Albino eats (*n* = 6)	500 mg/kg/day	14 days	Ellagic acid, gallotannins, anthocyanins, quercetin, kaempferol, luteolin glycosides, punicalin, punicafolin, luteolin, apigenin, anthocyanins, linoleic, oleic, palmitic, stearic, linolenic, and arachidic and palmitoleic acids	[515,516,517,518]
81. *Psidium guajava* L.	Guava	Fruit, leaves	Dysentery, diabetes and diarrhea	PPBG↓, FBG↓, HbA1c↓, IR↓, TG↓, TC↓, α-amylase and α-glucosidase activities↓, malondialdehyde↓	Prediabetes and mild T2DM patients (*n* = 120)	190 mg/kg 3 times a day	84 days	Quercetin, avicularin, apigenin, guaijaverin, kaempferol, hyperin, myricetin, gallic acid, catechin, epicatechin, chlorogenic acid, epigallocatechin gallate, caffeic acid	[519,520,521,522,523,524,525,526,527]
82. *Raphanus sativus* L.	Radish	Fruit, leaves	Gallbladder stone, jaundice, flatulence, indigestion, various gastric ailments, piles, constipation, indigestion, colic, dyspepsia, liver enlargement, diabetes	IR↓, intestinal glucose absorption↓, glucose uptake↑, glycoalbumin↓, fructosamine ↓	STZ-induced T2DM rats (*n* = 8)	2.2% of the diet/ day	21 days	Myricetin, catechin, epicatechin, quercetin, vanillic acid, sinapic acid, *p*-coumaric acid, β-carotene, camphene, piperitone, carvacrol, linoleic acid, oleic acid, anthocyanin	[528,529,530,531]
83. *Rosmarinus officinalis* L.	Rosemary	Leaves	Mycosis, alopecia, ultraviolet damage, skin cancer, inflammatory diseases, diabetes	FBG↓, TC↓, TG↓, LDL-C↓, GLUT-4 translocation↑, HDL↑, Irs1↓, IR↓, gluconeogenesis↓, glucose uptake↑	STZ-induced Wistar rats (*n* = 6)	4000 mg/kg/day	28 days	Flavonoids, carnosol, carnosoic, rosmarinic, ursolic, oleanolic, micromeric acids	[527,532,533,534,535,536,537]
84. *Rubus fruticosus*	Blackberry	Fruit, leaves	Mouthwash, gum inflammations, mouth ulcers, sore throat, respiratory disorders, anemia, diarrhea, dysentery, cystitis, diabetes, hemorrhoids	Blood glucose↓, α-amylase and α-glucosidase activities↓, oxidative stress↓	STZ-induced Sprague–Dawley rats (*n* = 6)	300 mg/L/day	35 days	Anthocyanins, malvidin, pelargonidin, cyanidins, kaempferol, quercetin, myricetin, *p*-coumaric acid, ferulic acid, rutin, coumarins, gallic acid	[538,539,540]
85. *Salvia hispanica* L.	Chia seeds	Seeds	Indigestion, hyperlipidemia, diabetes mellitus	Blood glucose↓, HbA1c↓, FBG↓, macrovascular complications↓, body weight↓, inflammatory cytokines↓, TC↓, TG↓, LDL-C↓, α-amylase and α-glucosidase activity↓	T2DM patients (*n* = 23)	40,000 mg/kg/day	84 days	Myricetin, quercetin, chlorogenic acid, kaempferol, and caffeic acid	[541,542,543,544,545]
86. *Sesamum indicum*	White sesame seeds	Seeds	Wound healing, amenorrhea, ulcer, asthma, hemorrhoids, inflammations, diabetes	Blood glucose↓, HbA1c↓, FBG↓, TC↓, PPBG↓, oxidative stress↓, IR↓ nephropathy↓	T2DM patients (18–60 years, *n* = 23)	30 mg/kg/day	90 days	Sesamin, sesaminol, gamma tocopherol, cephalin, flavonoids, phenolic acids, alkaloids, tannins, saponins, steroids, terpenoids	[546,547,548,549,550,551]
87. *Solanum lycopersicum* L.	Tomato	Fruit	Dermatitis, cancer, hypertension, hyperglycemia	Blood glucose↓ IR↓, SOD↑, GSH↑, MDA↓, inflammation↓	STZ-induced T2DM rats (*n* = 8)	30–270 mg/kg/day	56 days	Lycopene, carotenoids, homovanillic acid, chlorogenic acid, tomatine, kaempferol, quercetin, naringenin, *p*-coumaric acid, caffeic acid	[552,553,554,555,556,557,558,636]
88. *Solanum melongena*	Eggplant	Fruit, leaves	Arthritis, diabetes, dyslipidemia, bronchitis, asthma	Blood glucose↓, TC↓, TG↓, LDL-C↓, VLDL↓, HDL↑, oxidative stress↓, MDA↓, α-glucosidase activity↓, GLUT-4 translocation↑, glucose uptake↑, gluconeogenesis↓	Alloxan-induced diabetic rats (*n* = 6)	100–300 mg/kg/day	20 days	Solasodine, thiamin, niacin, chlorogenic acid, saponins, delphinidin, anthocyanin, phenols,	[559,560,561,562,563]
89. *Spinacia oleracea*	Spinach	Leaves	Remedy for bloody stools, diarrhea, stomachaches, obesity, diabetes	Retinopathy↓, MDA↓, inflammation↓, oxidative stress↓, AGEs formation↓, lipid peroxidation↓, IL-6↓, TNF-α↓, IR↓	STZ-induced Wistar rats (*n* = 10)	400 mg/kg/day	84 days	β-carotenoids, lutein, carotenoids, zeaxanthin, vitamins, minerals	[564,565,566,567,568,569]
90. *Syzygium aromaticum*	Clove	Flower buds	Flatulence, diarrhea, diabetes, indigestion	Blood glucose↓, PPAR-γ binding↑, aldose reductase↓	Diabetic KK-A^y^ Mice (*n* = 4)	657 mg/kg/day	21 days	Eugenol acetate, eugenol, gallic acid, terpenes, tannins, phenolics, steroids, flavonoids, glycosides, and saponins	[570,571,572,573,637]
91. *Syzygium cumini* L.	Java plum	Fruit, seeds, bark	Asthma, bronchitis, sore throat, biliousness, dysentery, diabetes, ulcers	Blood glucose↓, TG↓, TC↓, LDL↓, HDL↑, HMGCoA reductase↓, β cells function ↑, urine glucose↓	Alloxan- induced diabetic Albino rabbits (*n* = 5)	100 mg/kg/day	15 days	Anthocyanins, glucoside, isoquercetin, ellagic acid, kaemferol, myricetin	[574,575,576]
92. *Tamarindus indica* L.	Tamarind	Fruit, leaves, seeds	Inflammation, stomach pain, throat pain, rheumatism, wound, diarrhea, dysentery, fever, malaria, respiratory tract infection, constipation, cell cytotoxicity, gonorrhea, eye diseases	Blood glucose↓, body weight ↓, glucose tolerance ↑, β-cell function↑, glucose tolerance↑, β-cells regeneration↑	Alloxan-induced Wistar albino rats (*n* = 5)	100–250 mg/kg/day	14 days	Apigenin, anthocyanin, procyanidin, catechin, epicatechin, taxifolin, eriodyctiol, naringenin	[577,578,579,580,581]
93. *Theobroma cacao*	Cocoa	Fruit, husk, seeds	Measles, malaria, toothache as well as diabetes though improving insulin secretion, GLUT4 translocation, glucose uptake	Blood glucose↓, insulin secretion ↑, ATP↑, GSH↑, Nrf2↑ α-amylase and α-glucosidase activity ↓	INS-1 derived 832/13 rat insulinoma cell line	0.0025 mg/mL	24 h	Flavonoids, procyanidins, catechin, epicatechin, theobromine, caffeine	[582,583,584,585,586,587]
94. *Trichosanthes cucumerina* L.	Snake gourd	Fruit, leaves, seeds, roots	Bronchitis, headache, cathartic, anthelmintic, stomach disorders, indigestion, bilious fevers, boils, sores, eczema, dermatitis, psoriasis, ulcers, diabetes	FBG↓, IR↓, TC↓, TG↓, LDL-C↓, insulin secretion↑, intestinal glucose absorption ↓	STZ-induced Albino rats (*n* = 6)	750 mg/kg/day	28 days	Gallic acid, neochlorogenic acid, caffeic acid, *p*-coumaric acid, trans-ferulic acid, catechin hydrate, epicatechin, procyanidin A2, procyanidin B2, rutin, kaempferol, quercetin, ursolic acid, oleanolic acid	[588,589,590]
95. *Trigonella foenum-graecum*	Fenugreek seeds	Seeds	Ulcer, sinusitis, hay fever, diarrhea, diabetes, kidney diseases	Blood glucose↓, PPBG↓, FBG↓, glucose uptake↑, glucose tolerance↑, insulin sensitivity↑, intestinal glucose absorption↓	STZ-induced Long evans rats (*n* = 6)	500 mg/kg/day	28 days	Steroids, alkaloids, flavonoids, polyphenols, saponins	[591,592,593,594]
96. *Vaccinium corymbosum*	Blueberry	Fruit, leaves	Cold, inflammation, cardiovascular diseases, diabetes, ocular dysfunction	Blood glucose↓, IR↓, insulin secretion↑, retinopathy, α-amylase and α-glucosidase activities↓	STZ-induced Wistar rats (*n* = 8)	870 mg leaves/kg/day and 430 mg leaves + 1300 mg fresh fruits /kg/day	56 days	Anthocyanins, pectin, anthocyanidins, delphinidin, peonidin, malvidin, cyanidin, chlorogenic acid, malic acid, protocatechuic acid, petunidin	[595,596,597,638]
97. *Vigna radiata*	Mung bean	Seeds	Heat stroke, gastrointestinal disorders, dermatitis, hyperglycemia, hypertension, hyperlipidemia, melanogenesis	Blood glucose↓, TG↓, LDL↓, NO↓, α-amylase and α-glucosidase activity↓	Alloxan-induced Balb/c mice (*n* = 8)	200–100 mg/kg/day	10 days	Flavonoids, quercetin, myricetin, kaempferol, catechin, vitexin, isovitexin, coumaric acid luteolin, and caffeic and gallic acid	[598,599,600,601,602,603,604,639]
98. *Vitis vinifera* L.	Grapes	Fruit, seeds, peel	Diarrhea, hepatitis, stomachaches, varicose veins, hemorrhoids, atherosclerosis, diabetes, high blood pressure, heavy menstrual bleeding, uterine bleeding, constipation	Blood glucose↓, oxidative stress↓, β-cell regeneration↑, intestinal glucose absorption ↓	STZ-induced Wistar rats (*n* = 3)	250–500 mg/kg/day	15 days	Triterpenoid acids, oleanolic, betulinic acids, stilbenoid, gallic acid, catechin, epicatechin, gallocatechin, *p*-coumaric, and caffeic and ferulic acids	[605,606,607]
99. *Zea mays*	Corn	Grains, husk	Malaria, bladder stone, heart diseases, diabetes	body weight↓, FBG↓, IR↓, TC↓, TG↓, LDL-C↓, HDL↑, MDA↓, SOD↑, oxidative stress↓, α-amylase and α-glucosidase activity↓	STZ-induced HFF rats (*n* = 6)	300–1200 mg/kg/day	28 days	Flavonoids, alkaloids, saponins, phenols, tannins, phytosterols	[608,609,610,611,612,613]
100. *Zingiber officinale*	Ginger	Fruit	Muscular aches, pains, sore throats, cramps, constipation, indigestion, vomiting, arthritis, rheumatism, diabetes, sprains, hypertension, dementia, fever, infectious diseases, helminthiasis	Blood glucose↓, TC↓, TG, β-cell function↑, GLUT-4 activity↑, β-cell function↑, PPAR-γ↑, glucose uptake↑, creatinine↓, body weight↓, urea↓	STZ-induced Sprague Dawley rats (*n* = 8)	500 mg/kg/day	49 days	β-phellandrene, camphene, cineole, geraniol, curcumene, citral, terpineol, borneol, α-zingiberene, zingiberol, gingerols, shogaols 3-dihydroshogaols, paradols, dihydroparadols, gingerdiols, diarylheptanoids, isogingerol, isoshogaol gingerdiones	[614,615,616,640]

**Table 2 nutrients-16-03709-t002:** Phytoconstituents in dietary plants and their role in T2DM.

Dietary Plants	Plant Parts	Phytochemicals	Pharmacological Actions	Reference
1. *Abelmoschus esculentus* L.	Fruit, roots	Flavonoids, pectin, saponins, alkaloids	Lowers blood glucose and lipids, reduces insulin resistance, and enhances GLUT-4 translocation	[142,143,144]
2. *Actinidia chinensis*	Fruit	Triterpenoids, flavonoids, phenolic acids	Lowers serum glucose, inflammatory cytokines, blood lipids	[145,146,147,148]
3. *Aegle marmelos* L.	Fruit	Oleic acid, *p*-cymene, linolenic acid, retinoic acid, myristic acid	Enhances glucose tolerance and insulin sensitivity, suppresses α-amylase and α-glucosidase, delays intestinal glucose absorption	[149,150,151,641]
4. *Agaricus bisporus*	Rhizome	Catechin, lectin, β-glucans, Gallic acid, *p*-coumaric acid, Ferulic acid, Chlorogenic acid	Regulates insulin and glucagon secretion, reduces body weight and serum glucose	[152,153,154,155,156]
5. *Allium cepa*	Fruit	Quercetin, lectin, steroids, catechol, isoflavones, humulone, apigenin, rutin, myricetin, kaempferol, catechin	Decreases α-glucosidase activity, oxidative stress, boosts insulin and adiponectin secretion, protects β-cells	[157,158,159,160,161]
6. *Allium sativum* L.	Fruit	Allicin, β-resorcylic acid, gallic acid, rutin, protocatechuic acid, quercetin	Enhances insulin production, insulin secretion, glucose tolerance, insulin sensitivity, and GLUT-4 expression	[162,163,164,165,166,167]
7. *Aloe barbadensis Mill.*	Leaves	Flavonoids, proanthocyanidins, phenolic acids	Inhibits the glycation process, AGE formation and α-amylase, α-glucosidase enzyme activity	[166,168,169,170,171,172,173]
8. *Anacardium occidentale* L.	Nut, leaves, bark	Kaempferol, anacardic acid, quercetin, linolenic acid, gallic acid, myricetin, catechin, protocatechuic acid, epigallocatechin, naringenin, epicatechin	Inhibits glutamine-fructose-6-phosphate aminotransferase 1 (GFAT1) and dipeptidyl peptidase-4 (DPP-4) activity	[174,175,176,177,642]
9. *Ananas comosus* L.	Fruit, peel, leaves	Sinapic acid, daucosterol, coumarin, tannins, flavonoids, benzofuran, stillbenoid	Improves insulin sensitivity and body weight, inhibits HMGCoA reductase activity	[178,179,180,181,182,183,184]
10. *Apium graveolens*	Leaves, seeds, roots	Quercetin, thymoquinone, coumaric acid, gallic acid	Improves insulin sensitivity, GLUT-4 translocation, mitochondrial dysfunction, and inflammation	[185,186,187,188,189,190]
11. *Artocarpus heterophyllus*	Fruit, leaves, bark, seeds, roots	Carotenoid, tannins, sterols, Chysin, isoquercetine	Decreases postprandial glucose, blood lipids, and inhibits α-amylase and α-glucosidase	[193,194,195,196,197]
12. *Asparagus officinalis*	Stem	Asparagine, tyrosine, arginine, flavonoid, saponin, resin	Improves insulin secretion, insulin sensitivity, and β-cell function and lowers blood glucose	[198,199,200,201,202]
13. *Avena sativa*	Grains	β-glucan, oleic, linoleic acids, caffeic acids, coumaric acids, gallic acids, avenanthramides	Reduces glycosylated HbA1c, fasting blood glucose, postprandial glucose, insulin resistance	[203,204,205,206,207,643]
14. *Averrhoa carambola* L.	Fruit	Anthocyanins, rutin, triterpenoids, quercetin, catechin, epicatechin	Elevates insulin secretion, glucose uptake in skeletal muscles, and glycogen synthesis	[208,209,210,211,212]
15. *Azadirachta indica*	Leaves, stem, bark, flower, roots, fruit	Nimbidin, nimbin, nimbidol, quercetin, nimbosterone, ferulic acid, limonene, oleuropeoside	Inhibits α-glucosidase and glucokinase, stimulates insulin secretion	[213,214,215,216,217]
16. *Beta vulgaris*	Fruit	Lycopene, betalains, betagarin, betavulgarin, quercetin, kaempherol, betanins, carotenoid, coumarin	Inhibits α-amylase and α-glucosidase, gluconeogenesis, glycogenesis, and reduces serum glucose and lipids	[218,219,220]
17. *Brassica juncea*	Seeds	Chlorogenic acid, cinnamic acid, kaempferol, flavonoid, coumaric acid, vanillic acid	Improves blood glucose, glucose tolerance, insulin secretion and inhibits intestinal glucose absorption	[221,222,223,224]
18. *Brassica oleracea* var. capitata	Flower	Myricetin, quercetin, kaempferol, apigenin, luteolin, Anthocyanidin	Increases insulin sensitivity and β-cell function and lowers blood glucose	[225,226,227]
19. *Brassica oleracea* var. italica	Flower	Chlorogenic acid, apigenin, kaempferol, luteolin, quercetin and myricetin	Reduces ROS formation and oxidative stress, inhibits α-amylase and α-glucosidase, enhances insulin sensitivity and β-cell function	[228,229]
20. *Camellia sinensis*	Leaves	Theanine, proanthocyanidins, caffeine, myricetin, kaempferol, quercetin, chlorogenic acid, Catechins, epicatechin	Attenuates insulin resistance and oxidative stress, inhibits α-amylase and α-glucosidase, regulates inflammatory cytokines production	[230,231,232,233]
21. *Capsicum annuum* L.	Seeds	Flavonoids, carotenoids, flavones, apigenin, quercetin and isoquercetin	Activates AMPK, increases GLUT4 translocation and glucose uptake in skeletal muscle, and inhibits gluconeogenesis	[234,235,236,237,238]
22. *Carica papaya*	Fruit, seeds, leaves	Saponins, alkaloids, kaempferol, flavonoids, phenols, terpenoids, steroids, quercetin, caffeic acid	Decreases α-amylase and α-glucosidase activity, oxidative stress, and plasma blood glucose	[239,240,241,242]
23. *Carissa carandas*	Fruits	Lignans, flavonoids, Steroid, phenolic acid	Inhibits α-amylase and α-glucosidase, pro-inflammatory cytokine release, and lowers blood glucose	[243,244,245,246,247]
24. *Catharanthus roseus* L.	Flowers, leaves	Gallic acid, rutin, *p*-coumaric acid, caffeic acid, quercetin, kaempferol, chlorogenic acid, ellagic acid, coumarin	Increases insulin secretion and β-cell function, decreases blood glucose and lipids	[248,249,250,251,252]
25. *Centella asiatica*	Leaves	Centallase, quercetine, kaempferilm triterpene, ferulic acid	Decreases oxidative and inflammatory stress, body weight, serum glucose, and lipids	[253,254,255,256]
26. *Chenopodium quinoa*	Grains	Phytosterols, phytoecdysteroids, phenolics, tocophenols, betalains, tannins, glycine betaines	Inhibits α-glucosidase, improves insulin sensitivity, lowers postprandial glycemia	[257,258,259,260,261,262,263]
27. *Cicer arietinum*	Grains	Uridine, adenosine, tryptophan, 3-hydroxy-olean-ene, biochanin	Inhibits α-amylase, α-glucosidase, and dipeptidyl-4 (DPP4) enzymes	[264,265,266,267,268,269]
28. *Cinnamomum verum*	Bark	Cinnamaldehyde, cinnamates, cinnamic acid, eugenol, cinnamyl acetate, linalool	Enhances β-cell function, insulin secretion, and GLUT-4 translocation and inhibits α-amylase and α-glucosidase	[270,271,272,273]
29. *Citrullus lanatus* L.	Fruit, seeds	Lycopene, apigenin, kaempferol, rutin, *p*-coumaric acid, quercetin, ferulic acid	Inhibits α-amylase and α-glucosidase activity, enhances GLUT4 and GLUT2 translocation, and lowers blood glucose	[274,275,276,277]
30. *Citrus limon*	Fruit, peel, leaves	Limocitrin, D-limonene, hesperidin, naringenin, flavonoid	Decreases blood glucose and body weight and enhances GLUT4 translocation	[278,279,280,281,282,283,284,285]
31. *Citrus maxima*	Fruit, peel	Carotenoids, terpenoids, sterols, alkaloids, phenolics	Facilitates weight loss, inhibits α-amylase and α-glucosidase, increases glucose tolerance, and aids diabetic nephropathy	[286,287,288]
32. *Citrus reticulata*	Fruit, peel	Hesperidin, quercetin, flavonoids, tannins, anthraquinones	Enhances mRNA expression, GLUT-4 translocation, insulin sensitivity, and glucose tolerance	[289,290,291,292,293]
33. *Cocos nucifera*	Fruit, husk, water	Tannins, resins, flavonoid, alkaloids	Inhibits α-amylase and α-glucosidase activity, regenerates β-cells, and aids diabetic neuropathy	[295,296,297,298,299,300,301,302]
34. *Coffea Arabica* L.	Leaves, fruit, beans	Coffeasterin, caffeine, caffeic acid, *p*-coumaric acid, vanillic acid, ferulic acid, sitosterol, stigmasterol, kaempherol, quercetin, sinapic acid	Regenerates β-cells, inhibits α-glucosidase, and enhances insulin secretion	[303,304,305]
35. *Colocasia esculenta*	Stem, leaves	Viexin, isovitexin, orientin, isoorientin, rosmarinic acid, luteolin	Lowers blood glucose levels, oxidative stress, and inflammation, inhibits aldose reductase, and aids diabetic neuropathy	[306,307,308,309,310,644]
36. *Coriandrum sativum*	Seeds, leaves	Flavonoids, tocol, carotenoid, saponins	Inhibits TNF-α, IL-6, and AGE formation and aids diabetic neuropathy and nephropathy	[302,311,312,313,314]
37. *Crocus sativus* L.	Flower stigma	Safranal, β carotenes, crocetin, crocin, picrocrocin, zeaxanthene	Inhibits α-glucosidase and α-amylase, lowers blood glucose, lipids, and inflammatory cytokines	[315,316,317,318,319,320]
38. *Cuminum Cyminum* L.	Seeds	Cumin aldehyde, safranal, linalool, carvone, carvacrol	Protects β-cells, improves insulin secretion, lowers blood glucose	[321,322,323]
39. *Cucumis sativus*	Fruit, seeds	Cucurbitacin, cucumerin A and B, cucumegastigmanes I and II, orientin, apigenin	Reduces glucagon secretion, gluconeogenesis, and glycolysis, enhances insulin sensitivity	[324,325,326,327]
40. *Cucurbita pepo* L.	Fruit, seeds	β-carotene, lutein flavonoids, zeaxanthin, alkaloid	Lowers glucose in blood and urine, enhances glucose sensitivity and glutathione, reduces lipid levels	[328,329,330,331,332]
41. *Curcuma longa* L.	Fruit	Turmerine, turmerone, Cucurmin, curcuminol, demethoxycurcumin, caffeic acid, sinapic acid	Induces glucose uptake, GLUT-2 activity and insulin production, increases insulin secretion, insulin sensitivity, decreases insulin resistance	[333,334,335,336,337,645]
42. *Daucus carota*	Fruit	α and β-carotene, lutein, lycopene, anthocyanins, ascorbic acid	Regulates hyperglycemia, improves insulin resistance, delays intestinal glucose absorption, inhibits α-amylase and α-glucosidase	[338,339,340,341]
43. *Ficus carica*	Fruit, leaves, bark, roots	Eugenol, anthocyanins, phenolic acids, flavones, flavanols	Reduces postprandial glucose, plasma lipids, body weight, and β-cell apoptosis	[342,343,344,345,346,347]
44. *Fragaria ananassa*	Fruit, leaves	Quercetin, kaempferol, *p*-coumaric acid, *p*-tyrosol, methyl gallate, rutin	Ameliorates peripheral insulin resistance, inhibits α-amylase and α-glucosidase activity, increases insulin production	[263,348,349,350,351]
45. *Glycine max*	Seeds, leaves	β-conglycinin, flavonoids, saponins, phytosterols	Decreases insulin resistance, enhances glucose uptake in skeletal muscles through AMPK activation	[352,353,354,355,356]
46. *Helianthus* annuus	Flowers, seeds	Flavonoids, tocopherols, carotenoids, saponins, tannins, chlorogenic acid, caffeic acid	Reduces body weight, BMI, oxidative stress, AGE formation, and fasting blood glucose	[357,358,359,360]
47. *Hibiscus rosa-sinensis* Linn.	Flowers, leaves	Quercetin, cyanidin, ascorbic acid, genistic acid, lauric acid, thiamine, niacin	Stimulates β-cells, enhances insulin secretion and glycogen accumulation in the liver	[361,362,363,364,365]
48. *Hylocereus undatus*	Fruit, seeds	Oleic acid, gallic acid, lycopene, *p*-coumaric acid, linoleic acid, β-carotene	Attenuates plasma glucose, endothelial dysfunction, oxidative stress, and intestinal glucose absorption, and boosts insulin sensitivity	[366,367,368]
49. *Ipomoea batatas*	Fruit	Anthraquinones, coumarins, flavonoids, saponins, tannins, quercetin, chlorogenic acid, terpenoids	Mitigates insulin secretion and serum glucose, enhances β-cell function and insulin production	[369,370,371,372,373]
50. *Juglans regia* L.	Nut, leaves	Gallic acid, caffeoylquinic acid, coumaroylquinic, juglone, quercetin	Increases glucose uptake and inhibits α-glucosidase, α-amylase, and protein tyrosine phosphatase 1B (PTP1B) activity	[374,375,376,377,646]
51. *Lactuca sativa*	Leaves	Flavonoids, quercetin, flavonols, anthocyanins, lutein, β-carotene	Inhibits α-amylase, α-glucosidase, and DPP-4, improves postprandial glucose and blood lipids	[378,379,380,381,382]
52. *Lagenaria siceraria*	Fruit, leaves, seeds	cucurbitacin B, cucurbitacin D, cucurbitacin E, isoquercitrin, kaempferol, gallic acid	Improves glucose tolerance and insulin production and inhibits intestinal glucose absorption	[383,384,385,386]
53. *Laurus nobilis*	Leaves	Eugenol, kaempferol, syringic acid, quercetin, apigenin, luteolin	Enhances β-cell function and insulin sensitivity and inhibits α-amylase and α-glucosidase	[387,388,389,390]
54. *Litchi chinensis*	Fruit, seeds	Sterols, triterpenoids, flavonoids, phenolics	Improves insulin resistance, serum triglyceride levels, and glucose tolerance and inhibits α-glucosidase activity	[391,392,393]
55. *Luffa acutangula*	Fruit, seeds	Apigenin, luteolin, myristic acid, α-pinene, carotene, oleanolic acid, β-myrcene, linalool	Enhances insulin secretion, suppresses glycogenolysis and gluconeogenesis	[394,395]
56. *Malus domestica Borkh*	Fruit, peel	Quercetin, pectin, flavonols, flavanols, catechin epicatechin, cyanidin galactoside	Improves endothelial function, lipid homeostasis, insulin resistance, and lowers serum glucose	[396,397,398,399,400,401,402,403,404,405]
57. *Mangifera indica*	Fruit, peel, bark, seeds	Mangiferin, rhamnetin, catechin, epicatechin, gallic acid	Increases insulin sensitivity, lowers postprandial glucose, inhibits α-amylase and α-glucosidase	[406,407,408,409,410,411]
58. *Mentha spicata*	Leaves	Limonene, carvone, linalool, piperitone	Suppresses α-amylase and α-glucosidase activity and oxidative stress, and decreases blood glucose and lipids	[412,413,414]
59. *Moringa oleifera* Lam.	Fruit, leaves	Anthocyanins, sitogluside, tannin, anthraquinones, β-carotene	Inhibits α-amylase and α-glucosidase, lowers postprandial glucose and cholesterol, and improves lipid metabolism	[415,416,417,418,419,647]
60. *Momordica charantia*	Fruit, leaves, seeds	Triterpene, proteid, steroids, flavonoids, ascorbic acid, saponins	Regenerates β-cells, increases glucose uptake in skeletal muscle, and suppresses intestinal glucose absorption	[420,421,422,423,424,425,426,427]
61. *Morus alba* L.	Fruit, leaves	Quercetin, isoquercetin, stillbenoids, flavonoids	Enhances insulin secretion, lowers blood glucose and blood lipids, and promotes GLUT-4 translocation	[429,430,431,432]
62. *Murraya koenigii* L.	Leaves	Murrayanol, mahanimbine, kaemferol, catechin, apgenin	Regenerates β-cells, inhibits α-amylase and α-glucosidase, lowers blood glucose	[433,434,435,436]
63. *Myristica fragrans* Houtt.	Fruit, seeds	Lignan, flavonoids, terpenes, coumarin	Inhibits TNF-α and IL-6 release, ameliorates blood glucose, β-cell function, inflammation, and obesity	[437,438,439,440]
64. *Nigella sativa* L.	Seeds	Thymoquinone, thymol, limonene, carvacrol, *p*-cymene, linoleic acid, oleic acid	Inhibits hepatic gluconeogenesis, α-amylase, and α-glucosidase, increases insulin sensitivity	[441,442,443,444,445]
65. *Ocimum sanctum* L.	Leaves, seeds	Ursolic acid, eugenol, carvacrol, linalool, caryophyllene	Lowers serum glucose and albumin, increases insulin secretion and lipid metabolism, regenerates β-cells	[446,447,448,449,648]
66. *Olea europaea* L.	Fruit, leaves	Secoiridoid glycoside, oleuropein, oleanolic acid, flavonoid, cinnamic acid	Enhances glucose tolerance, reduces body weight, inhibits gluconeogenesis, and lowers plasma glucose	[450,451,452,453,454,455]
67. *Origanum vulgare*	Leaves	Rosmarinic acid, apigenin, luteolin	Increases glucose uptake in skeletal muscle and GLUT-2, decreases blood glucose and oxidative stress, inhibits α-amylase and α-glucosidase	[456,457,458,459]
68. *Passiflora edulis*	Fruit, peel	Piceatannol, flavonoids, tocopherols, carotenoid, gallic acid, rutin	Improves serum glucose, insulin sensitivity, glucose tolerance, and glucose uptake in skeletal muscle, and reduces lipid accumulation and body weight	[460,461,462,463,464,465,466,467,468]
69. *Persea americana Mill.*	Fruit, leaves, seeds, bark	Myricetin, luteolin, gallic acid, ascorbic acid	Activates PI3K to facilitate insulin action, inhibits α-amylase and α-glucosidase	[469,470,471,472,473]
70. *Petroselinum crispum*	Leaves, seeds, roots	Coumarins, tocopherols, apigenin, myristicin	Regulates plasma glucose, body weight, and glutathione levels, increases glucose uptake in skeletal muscles, and inhibits gluconeogenesis	[474,475,476,477]
71. *Phaseolus vulgaris* L.	Seeds	*p*-coumaric acid, myricetin, naringenin, gallic acid, quercetin, catechin, kaempferol, ferulic acid	Suppresses α-glucosidase activity and gluconeogenesis, delays the absorption of glucose, increases insulin sensitivity	[478,479,480,481]
72. *Phoenix dactylifera* L.	Fruit, leaves	Flavonoids, oleic acid, linoleic acid, catechin, epicatechin, apigenin, naringenin, anthocyanin	Enhances β-cell function and insulin secretion, decreases blood glucose, inhibits α-amylase and α-glucosidase	[482,483,484,485]
73. *Phyllanthus emblica* L.	Fruit, leaves, bark, roots	Gallic acid, ellagic acid, pectin, quercetin, linoleic, oleic acid, myristic acid,	Inhibits α-amylase and α-glucosidase, activates AMPK, and lowers blood glucose	[486,487,488,489]
74. *Piper betle* L.	Leaves	Eugenol, selinene, hydroxychavicol, cadinene, caryophyllene	Elevates insulin production and glucose usage, activates glucokinase, and lowers plasma glucose	[490,491,492,493]
75. *Pisum sativum* L.	Seeds	Uridine, adenosine, tryptophan, 3-hydroxy-olean-ene, biochanin	Inhibits α-amylase, α-glucosidase, and dipeptidyl-4 (DPP4) enzymes	[494,495,496,497]
76. *Prunus armeniaca* L.	Fruit, leaves	Quercetin, ferulic acid, chlorogenic acid, lutein, catechin, epicatechin	Stimulates insulin secretion, decreases oxidative stress, inhibits α-amylase and α-glucosidase	[498,499,500]
77. *Prunus domestica*	Fruit	Catechin, epicatechin, chlorogenic acid, kaempferol, quercetin	Inhibits HMGCoA reductase and α-amylase, lowers blood glucose, lipids, and oxidative stress	[501,502,503,504,505,506,507,508]
78. *Prunus dulcis*	Nut	Oleic acid, linoleic acid, P-coumaric acid, anthocyanin, kaempferol, quercetin, chlorogenic acid	Increases insulin production and decreases stomach emptying time	[509,510]
79. *Prunus persica* L.	Fruit, peel, leaves	Naringenin, ferulic acid, Chlorogenic acid, astragalin, carotenoid, anthocyanin, caffeic acid	Ameliorates insulin secretion, pancreatic β-cell regeneration, and inhibits α-amylase and α-glucosidase	[511,512,513,514]
80. *Punica granatum*	Fruit, peel, seeds	Punicalin, punicsfolin, apigenin, quercetin, ellagic acid, gallotannins, anthocyanins, luteolin, kaempferol, lycopene	Enhances insulin sensitivity, insulin production, GLUT-4 translocation, and lowers blood glucose	[515,516,517,518]
81. *Psidium guajava* L.	Fruit, leaves	Quercetin, avicularin, guaijaverin, tannins, triterpenes	Decreases plasma glucose, gluconeogenesis, triglycerides, total cholesterol, and increases glucose uptake in skeletal muscle	[519,520,521,522,523,524,525,526,527,649]
82. *Raphanus sativus* L.	Fruit, leaves	Myricetin, catechin, epicatechin, quercetin, vanillic acid, Oleic acid, *p*-coumaric acid, β-carotene	Inhibits intestinal glucose absorption, increases glucose uptake in skeletal muscle, and lowers blood glucose	[528,529,530,531]
83. *Rosmarinus officinalis* L.	Leaves	Rosmarinic acid, ursolic acid, oleonic acid, carnosol	Enhances insulin sensitivity, GLUT-4 translocation, glucose uptake in skeletal muscle, and inhibits gluconeogenesis	[527,532,533,534,535,536,537]
84. *Rubus fruticosus*	Fruit, leaves	anthocyanins, malvidins, pelargonidin, cyanidins, kaempferol, quercetin	Lowers blood glucose, inhibits α-amylase and α-glucosidase	[538,539,540]
85. *Salvia hispanica* L.	Seeds	Omega-3 fatty acid, myricetin, quercetin, chlorogenic acid, kaempferol, caffeic acid	Inhibits α-amylase and α-glucosidase, reduces body weight, inflammatory cytokine release, and blood glucose and lipids	[541,542,543,544,545]
86. *Sesamum indicum*	Seeds	Sesamin, sesaminol, tocopherol, flavonoids, saponins, steroids, terpenoids	Attenuates postprandial glucose and oxidative stress, improves insulin secretion, glutathione levels, and lipid metabolism	[546,547,548,549,550,551]
87. *Solanum lycopersicum* L.	Fruit	Lycopene, tomatine, kaempferol, quercetin, chlorogenic acid, β-carotene, naringenin	Attenuates plasma glucose, inflammation, and insulin resistance via PI3K/Akt, FOXO1, and PPAR-γ regulation	[552,553,554,555,556,557,558]
88. *Solanum melongena*	Fruit, leaves	Thiamin, niacin, flavonoids, saponins, tannins, triterpenoids, anthraquinones	Enhances glucose uptake in skeletal muscles, GLUT-4 translocation, reduces gluconeogenesis, α-amylase and α-glucosidase enzymes, and hyperlipidemia	[559,560,561,562,563]
89. *Spinacia oleracea*	Leaves	β-carotenoids, lutein, carotenoids, zeaxanthin	Reduces serum C-reactive protein, TNF α, IL-6, and excess AGE production, and aids in retinopathy	[564,565,566,567,568,569]
90. *Syzygium aromaticum*	Flower buds	Eugenol, gallic acid, ferulic acid, catechin, quercetin	Inhibits α-amylase, α-glucosidase, and aldose reductase, lowers blood glucose, and activates PPAR-γ	[570,571,572,573]
91. *Syzygium cumini* L.	Fruit, seeds, bark	Anthocyanins, isoquercetin, ellagic acid, kaempferols, myricetin	Regenerates β-cells, improves insulin production, and lowers glucose in plasma and urine	[574,575,576,650]
92. *Tamarindus indica* L.	Fruit, leaves, seeds	Catechin, anthocyanin, epicatechin, apigenin	Lowers blood glucose, inhibits α-amylase and α-glucosidase, elevates glucose tolerance, and regenerate β-cells	[577,578,579,580,581]
93. *Theobroma cacao*	Fruit, husk, seeds	Catechin, epicatechin, procyanidin, saponins, terpenoids	Protects β-cells, inhibits α-amylase and α-glucosidase, elevates ATP, GSH, Nrf2, and glucose uptake in skeletal muscle	[582,583,584,585,586,587]
94. *Trichosanthes cucumerina* L.	Fruit, leaves, seeds, roots	Carotenoids, gallic acid, neochlorogenic acid, caffeic acid, *p*-coumaric acid, rutin, kaempferol, quercetin, ursolic, oleanolic acids	Stimulates insulin secretion, enhances the peripheral use of glucose, and prevents intestinal glucose absorption	[588,589,590]
95. *Trigonella foenum-graecum*	Seeds	Steroids, alkaloids, flavonoids, polyphenols, saponins	Decreases blood glucose and enhances glucose uptake, insulin sensitivity, and glucose tolerance	[591,592,593,594]
96. *Vaccinium corymbosum*	Fruit, leaves	Anthocyanins, pectin, anthocyanidins, delphinidin, peonidin, malvidins	Suppresses α-amylase and α-glucosidase activity and aids diabetic retinopathy	[595,596,597]
97. *Vigna radiata*	Seeds	quercetin, myricetin, kaempferol, catechin, coumaric acid, luteolin, caffeic, gallic acid	Hinders gluconeogenesis and glycolysis, inhibits α-glucosidase and α-amylase	[598,599,600,601,602,603,604]
98. *Vitis vinifera* L.	Fruit, seeds, peel	Catechin, epicatechin, epicatechin gallate, quercetin, myricetin, resveratrol	Regenerates β-cells, lowers blood glucose, inhibits intestinal glucose absorption, and facilitates glycogen synthesis	[605,606,607,651]
99. *Zea mays*	Grains, husk	Hirsutrin, flavonoids, alkaloids, saponins, phenols, tannins, phytosterols	Ameliorates diabetic complications by suppressing aldose reductase and reducing galactitol formation, inhibits α-amylase and α-glucosidase activity	[608,609,610,611,612,613]
100. *Zingiber officinale*	Fruit	Vanilloids, gingerol, paradol, shogaols, zingerone, gingerdiols,	Activates GLUT-4 and PPAR-γ, protects β-cells, facilitates glucose uptake in tissues	[614,615]

## 7. Conclusions and Future Perspectives

Plant-based dietary adjuncts represent a promising natural approach for the management of T2DM due to the vast array of phytochemicals they contain. Unlike conventional medications, such natural products are widely accessible, affordable, and generally free from adverse effects. Integrating plant-derived foods into the daily diet not only helps control the hyperglycemia observed in DM but also supports weight management in obese individuals and has broad health benefits [652,653,654]. The plants highlighted in this review can interact in a variety of ways to regulate blood glucose and restore insulin sensitivity. In addition, it is important to mention that fiber-rich plants also play a role in obesity management [655,656,657]. To date, the majority of scientific studies on antidiabetic plants have been carried out in vitro and/or in vivo. More research is needed to identify the antidiabetic potential of the plants selected in this review in patients with diabetes. Furthermore, more research is needed to better understand the identity and mechanism of action of the active phytoconstituents at the molecular level. We also need to determine what the future holds for the potential exploitation of these natural products for the development of new and safer pharmaceuticals that could assist the treatment of DM and its complications.

## Figures and Tables

**Figure 1 nutrients-16-03709-f001:**
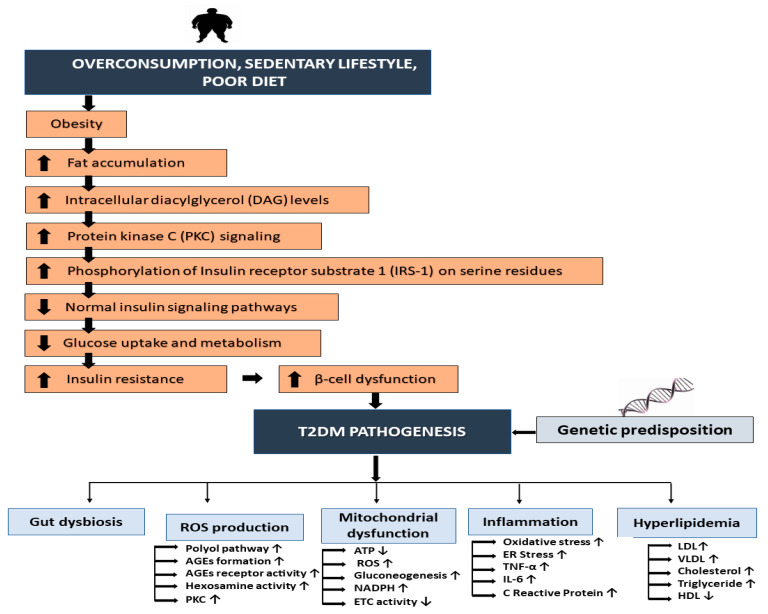
Mechanistic overview of T2DM pathogenesis. Obesity, driven by overconsumption, a sedentary lifestyle, and poor diet, results in increased fat accumulation and intracellular diacylglycerol (DAG) levels, which activate Protein Kinase C (PKC) signaling. This leads to the phosphorylation of Insulin Receptor Substrate 1 (IRS-1) on serine residues, impairing normal insulin signaling pathways, reducing glucose uptake, and increasing insulin resistance. Insulin resistance, in turn, promotes β-cell dysfunction, contributing to the progression of type 2 diabetes mellitus (T2DM). Genetic predisposition further influences the disease’s development. Key processes downstream include gut dysbiosis, reactive oxygen species (ROS) production, mitochondrial dysfunction, inflammation, and hyperlipidemia. Increase (↑) and decrease (↓).

**Figure 2 nutrients-16-03709-f002:**
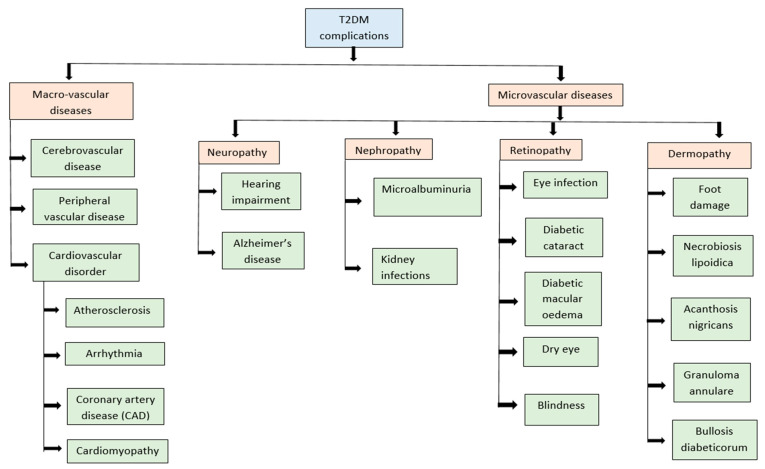
Flow chart of T2DM-associated vascular complications.

**Figure 3 nutrients-16-03709-f003:**
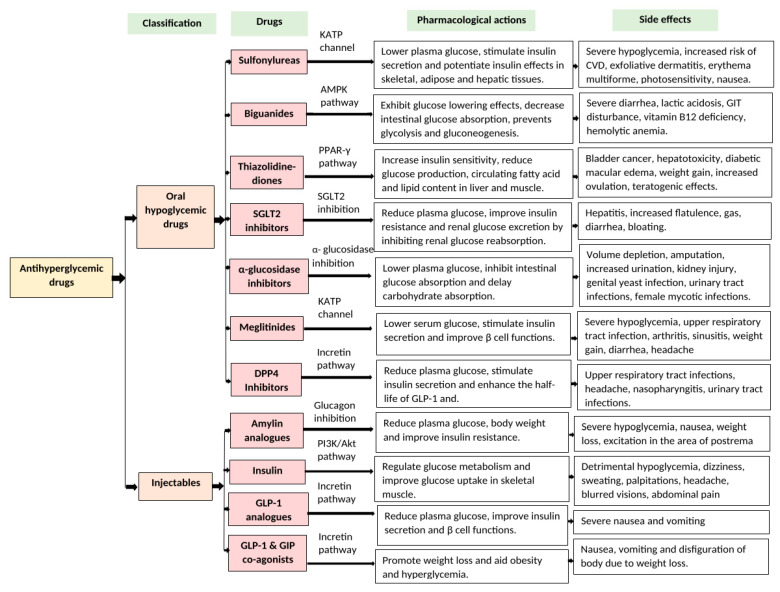
Flow chart of the current oral and injectable antidiabetic drugs, their pharmacological actions, and adverse side effects.

**Figure 4 nutrients-16-03709-f004:**
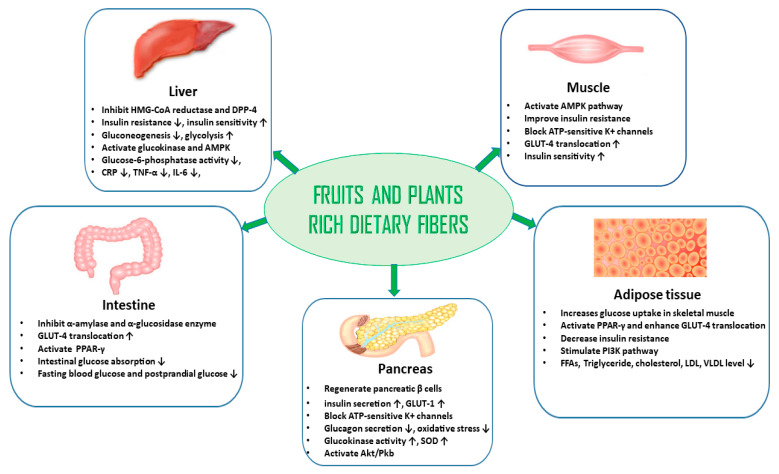
Antidiabetic effects of dietary fiber-rich plants and fruits on various organs and tissues. Dietary fiber-rich herbs and fruits exhibit antihyperglycemic properties by activating several molecular pathways. They may contribute to the regeneration of pancreatic β-cells; increase insulin secretion; enhance insulin sensitivity; increase glucose uptake in tissues; enhance GLUT-4 translocation; increase glycolysis in the liver; activate the AMPK, PPAR-γ, Akt/Pkb, or PI3K pathways in adipose tissue; improve glucokinase activity; reduce insulin resistance; delay intestinal glucose absorption; lower fasting blood sugar and postprandial glucose; reduce glucagon secretion and oxidative stress; inhibit α-amylase, α-glucosidase, DPP-4, and glucose-6-phosphatase enzymatic activity; decrease gluconeogenesis; suppress TNF-α and IL-6 release; and block ATP-sensitive K^+^ channels in the pancreas and muscle to regulate blood glucose levels. Increase (↑) and decrease (↓).

**Figure 5 nutrients-16-03709-f005:**
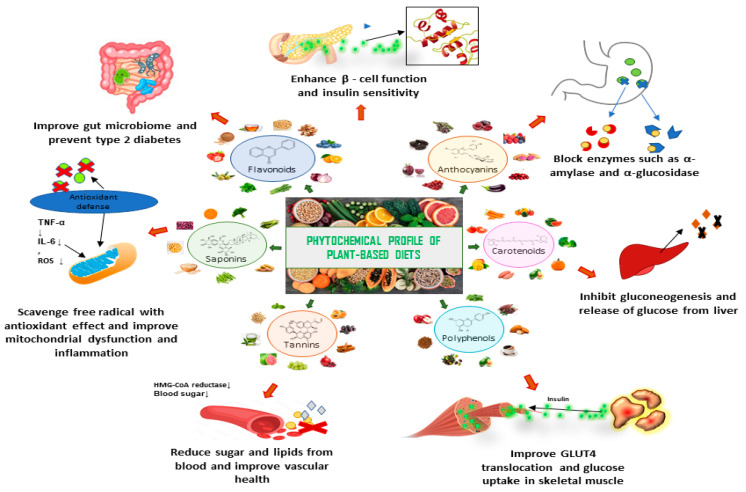
Antidiabetic effects of phytochemicals in diabetes management. Phytochemicals, including flavonoids, anthocyanins, carotenoids, saponins, tannins, and polyphenols, are abundant in plant-based foods such as fruits, vegetables, legumes, and whole grains. These compounds play a critical role in diabetes management by (i) enhancing beta-cell function, (ii) improving insulin sensitivity, (iii) inhibiting carbohydrate-digesting enzymes such as α-amylase and α-glucosidase, (iv) reducing glucose production in the liver, (v) improving gut health and promoting beneficial microbiota, and (vi) reducing inflammation and oxidative stress. Such actions contribute to improved glycemic control, thereby limiting the impact of diabetes and its complications. Increase (↑) and decrease (↓).

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
