# Peer review of "Plant-Based Diets and Phytochemicals in the Management of Diabetes Mellitus and Prevention of Its Complications: A Review"

_nutrients, 2024, doi:10.3390/nu16213709_

Round 1
Reviewer 1 Report
Comments and Suggestions for Authors
The authors present a review article that aimed to provide an overview of the pathogenesis and current therapeutic management of DM, with a particular focus on the promising potential of plant-based foods.
I would like to raise the following concerns.
1.
It seems that Figures 1-5 in the review might focus on well-established aspects of diabetes, which may not fully align with the main theme of this review article on "Plant-Based Diets and Phytochemicals in the Management of Type 2 Diabetes Mellitus and Prevention of its Complications."
Would you like to revise these Figures to emphasize the novel aspects of plant-based diets and phytochemicals in diabetes management, or perhaps include new Figures that focus on this theme?
2.
It seems that Table 1 on "Traditional uses, pharmacological actions, and phytoconstituents of dietary plants" may lack critical research design details, such as epidemiological study terms (study design types) and effect sizes, as well as the statistical significance of findings (confidence intervals and p-values).
Including these elements would strengthen the scientific rigor of the Table and provide more context on the effectiveness and relevance of these dietary plants in managing T2DM.
3.
A concise visual representation, such as a Figure or summary Table, could effectively clarify the role of plant-based diets and phytochemicals in the management of Type 2 Diabetes Mellitus and the prevention of its complications.
4.
The review article is lengthy and difficult to read. It is suggested to condense the article to improve readability.
Reviewer 2 Report
Comments and Suggestions for Authors
Review of "Plant-Based Diets and Phytochemicals in the Management of Diabetes Mellitus and Prevention of its Complications: A Review" (nutrients-3196408).
This review article focuses on the effect of plant-based phytochemicals on diabetes. The authors showed the association between one hundred plant-based diet, edible plant, and dietary supplement and diabetes. The author's efforts are commendable. However, this reviewer has several questions and comments.
1. The main purpose of this review was to show the effect of plant-based diet, edible plant, and dietary supplement on diabetes, especially type 2 diabetes. Thus, explanation of types of diabetes and antidiabetic drugs were not necessary. Furthermore, explanation of Figure 1 was not necessary. On the other hand, the authors should be shown the association between plant-based diets and phytochemicals.
2. Methodology. If this study was a narrative review, this part also unnecessary. On the other hand, if this was a systematic review, there are some missing evaluation items that need to be added.
3. In this review, 100 plant-based diet, edible plant, and dietary supplement are listed, but how can this reviewer determine if this is all of them?
4. How much of each phytochemical is contained in each plant-based diet, edible plant, and dietary supplement. It is not clear what role each Phytochemicals plays for diabetes.
5. Table 1. Several references are shown for each plant-based diet, edible plant, and dietary supplement, but it is puzzling that only one study is shown in column.
6. The authors should also summarize through what molecular mechanism each plant-based diet, edible plant, and dietary supplement or each phytochemical act on diabetes in Table
Reviewer 3 Report
Comments and Suggestions for Authors
Comments to authors
The authors presented a review, apparently narrative (although it is not specified, as in the search they say systematic search), on the effect of 100 potentially useful plants in diabetes.
· Line 90: It says a systematic search was done, but systematic searches are usually done in systematic reviews, with a much more precise structure and methodology. I suggest changing systematic search to exhaustive search.
· The authors say that 1,500 articles were initially identified, of which 671 were included in the final analysis. The authors used 670 references, which I assume is what they mean. However, the search is done to find articles of interest to answer the objective of the review (which is plants and diabetes, i.e. the references for section 6).
· Table 1 could be organised by type of evidence, i.e. whether the evidence is based on preclinical models (animals, cell lines, etc.) or on humans. That is, add a row and say "evidence in preclinical studies", and below that add all the rows for the plants that are based on preclinical studies, and then a row that says "evidence in humans", and add the rows that correspond to studies in humans.
Round 2
Reviewer 1 Report
Comments and Suggestions for Authors
No further comment
Author Response
Thank you for your positive feedback and acceptance of our revised manuscript.
Reviewer 2 Report
Comments and Suggestions for Authors
This reviewer still thinks there is a problem.
Resopnse to Comment 1: "We feel that it is important to lay down a broad background on diabetes for the reader and carefully consider the various classes of antidiabetic drugs currently available, especially given the major advances since use of only biguanides and sulphonylureas was common place."
However, since the readers of this article are expected to have some knowledge of diabetes, I believe that a detailed drug description is unnecessary. Furthermore, I do not think a detailed explanation of complications is necessary.
Resopnse to Comment 5. The references cited in Table were not appropriate. Which reference represents which content should be described in detail.
